# Dynamic Visual Reasoning by Learning Differentiable Physics Models from Video and Language

**Mingyu Ding**
MIT CSAIL and HKU

**Zhenfang Chen**
MIT-IBM Watson AI Lab

**Tao Du**
MIT CSAIL

**Ping Luo**
HKU

**Joshua B. Tenenbaum**
MIT BCS, CBMM, CSAIL

**Chuang Gan**
MIT-IBM Watson AI Lab

## Abstract

In this work, we propose a unified framework, called Visual Reasoning with Differentiable Physics (VRDP) [1], that can jointly learn visual concepts and infer physics models of objects and their interactions from videos and language. This is achieved by seamlessly integrating three components: a visual perception module, a concept learner, and a differentiable physics engine. The visual perception module parses each video frame into object-centric trajectories and represents them as latent scene representations. The concept learner grounds visual concepts (*e.g.*, color, shape, and material) from these object-centric representations based on the language, thus providing prior knowledge for the physics engine. The differentiable physics model, implemented as an impulse-based differentiable rigid-body simulator, performs differentiable physical simulation based on the grounded concepts to infer physical properties, such as mass, restitution, and velocity, by fitting the simulated trajectories into the video observations. Consequently, these learned concepts and physical models can explain what we have seen and imagine what is about to happen in future and counterfactual scenarios. Integrating differentiable physics into the dynamic reasoning framework offers several appealing benefits. More accurate dynamics prediction in learned physics models enables state-of-the-art performance on both synthetic and real-world benchmarks while still maintaining high transparency and interpretability; most notably, VRDP improves the accuracy of predictive and counterfactual questions by 4.5% and 11.5% compared to its best counterpart. VRDP is also highly data-efficient: physical parameters can be optimized from very few videos, and even a single video can be sufficient. Finally, with all physical parameters inferred, VRDP can quickly learn new concepts from few examples.

## 1  Introduction

Dynamic visual reasoning about objects, relations, and physics is essential for human intelligence. Given a raw video, humans can easily use their common sense of intuitive physics to explain what has happened, predict what will happen next, and infer what would happen in counterfactual situations. Such human-like physical scene understanding capabilities are also of great importance in practical applications such as industrial robot control [2, 53].

---

[1]Project page: http://vrdp.csail.mit.edu/

35th Conference on Neural Information Processing Systems (NeurIPS 2021).

Previous works have made great efforts to build artificial intelligence (AI) models with such physical reasoning capabilities. One popular strategy is to develop pure neural-network-based models [63, 19, 40]. These methods typically leverage end-to-end neural networks [32, 35] with powerful attention modules such as Transformer [69, 21] to extract attended features from both video frames and question words, based on which they answer questions directly. Despite their high question-answering accuracy on CLEVRER [79], a challenging dynamic visual question-answering benchmark, these black-box models neither learn concepts nor model objects' dynamics. Therefore, they lack transparency, interpretability, and generalizability to new concepts and scenarios. Another common approach to dynamic visual reasoning is to build graph neural networks (GNNs) [47] to capture the dynamics of the scenes. These GNN models [54, 79, 16] treat objects in the video as nodes and perform object- and relation-centric updates to predict objects' dynamics in future or counterfactual scenes. Such systems achieve decent performance with good interpretability on CLEVRER by combining the GNN-based dynamics models with neural-symbolic execution [58, 80]. However, these dynamic models do not explicitly consider laws of physics or use concepts encoded in the question-answer pairs associated with the videos. As a result, they show limitations in counterfactual situations that require long-term dynamics prediction.

Although (graph-)neural-network-based approaches have achieved competitive performance on CLEVRER, dynamic visual reasoning is still far from being solved perfectly. In particular, due to the lack of explicit physics models, existing models [79, 19, 16] typically struggle to reason about future and counterfactual events, especially when training data is limited. For this reason, one appealing alternative is to develop explicit physics-based methods to model and reason about dynamics, as highlighted in the recent development of differentiable physics engines [9, 17, 68, 18, 66] and their applications in robotics [9, 17, 68]. However, these physics engines typically take as input a full description of the scene (*e.g.*, the number of objects and their shapes) which usually requires certain human priors, limiting their availability to applications with well-defined inputs only.

In this work, we take an approach fundamentally different from either network-based methods or physics-based methods. Noting that deep learning based methods excel at parsing objects and learning concepts from videos and language, and physics laws are good at capturing object dynamics, we propose Visual Reasoning with Differentiable Physics (VRDP), a unified framework that combines a visual perception module, a concept learner, and a differentiable physics engine. VRDP jointly learns object trajectories, language concepts, and objects' physics models to make accurate dynamic predictions. It starts with a perception module running an object detector [31] on individual frames to generate object proposals and connect them into trajectories based on a motion heuristic. Then, a concept learner learns object- and event-based concepts, such as 'shape', 'moving', and 'collision' as in DCL [16, 58]. Based on the obtained object trajectories and attributes, the differentiable physics engine estimates all dynamic and physical properties (*e.g.*, velocity, angular velocity, restitution, mass, and the coefficient of resistance) by comparing the simulated trajectories with the video observations. With these explicit physical parameters, the physics engine reruns the simulation to reason about future motion and causal events, which a program executor then executes to get the answer. The three components of VRDP cooperate seamlessly: the concept learner grounds physical concepts needed by the physics engine like 'shape' onto the objects detected by the perception module; the differentiable physics engine estimates all physical parameters and simulates accurate object trajectories, which in turn help the concept learning process in the concept learner.

Compared with existing methods, VRDP has several advantages thanks to its carefully modularized design. First, it achieves the state-of-the-art performance on both synthetic videos (CLEVRER [79]) and real-world videos (Real-Billiard [63]) without sacrificing transparency or interpretability, especially in situations that require long-term dynamics prediction. Second, it has high data efficiency thanks to the differentiable physics engine and symbolic representation. Third, it shows strong generalization capabilities and can capture new concepts with only a few examples.

## 2  Related Work

**Visual Reasoning**  Our model is related to reasoning on vision and natural language. Existing works can be generally categorized into two streams as end-to-end approaches [40, 83, 74, 41, 5] and neuro-symbolic approaches [80, 29, 30, 24, 4, 59, 45, 58, 36, 3]. The end-to-end methods [40, 83, 74, 60] typically tackle the visual question answering (VQA) problem by designing monolithic multi-modal deep networks [32, 35]. They directly output answers without explicit and interpretable mechanisms.

Beyond question answering, neuro-symbolic methods [80, 29, 59, 45, 58] propose a set of visual-reasoning primitives, which manifest as an attention mechanism capable of performing complex reasoning tasks in an explicitly interpretable manner.

Dynamic visual reasoning in videos has attracted much research attention. Many video question answering datasets [27, 49, 42, 67, 81] and the methods [82, 51, 38, 75, 78, 22] built on them mainly focus on understanding diverse visual scenes, such as human actions (MovieQA [67]) or 3D object movements without physical and language cues (CATER [27]). Differently, CLEVRER [79] targets the physical and causal relations grounded in dynamic videos of rigid-body collisions and asks a range of questions that requires the modeling of long-term dynamic predictions. For this reason, we evaluate our method and compare it with other state-of-the-arts on CLEVRER.

Both end-to-end [63, 19] and neuro-symbolic methods [79, 16] have been explored on CLEVRER. However, they either lack transparency or struggle for long-term dynamic prediction. In this paper, we perform high-performance and interpretable reasoning by recovering the physics model of objects and their interactions (*e.g.*, collisions) from visual perception and language concepts.

**Physical Models** Physical models are widely used in video prediction [50, 23, 76, 77], neural simulation and rendering [52, 54], and dynamic reasoning [10, 71]. For example, PhysNet and its variants [50, 23, 76, 77] leverage global or object-centric deep features to predict the physical motion in video frames. Some other related works [61, 1, 65, 46] extend physical models to predict the effect of forces and infer the geometric attributes and topological relations of cuboids.

In this work, we focus on dynamic visual reasoning about object interactions, dynamics, and physics with question answering, which is central to human intelligence and a key goal of artificial intelligence. Solving such tasks requires a good representation and understanding of physics models. A common choice is to train a deep neural network for physical property estimation (*e.g.*, location and velocity) based on learned visual and dynamic priors [15, 10, 71, 70, 43, 54, 55, 34, 33]. However, since these neural networks do not model physics laws, generalizing them to unseen events or counterfactual scenarios could result in unexpected results. Our work is different and more physics-based: Inspired by the recent advances in differentiable physics [9, 17, 68, 18, 66, 37], we implement an impulse-based differentiable rigid-body simulator and leverage the power of its gradients to infer dynamics information about the scene.

**Physical Scene Understanding** Our work is also relevant to studies on physical scene understanding [7, 73, 66, 64, 8, 25, 26, 20, 50, 39], most of which propose pure neural-network solutions without explicitly incorporating physics models. Benchmarks like PHYRE [7] study physical understanding and reasoning based on pure videos without concept learning and language inference. Based on such benchmarks, some works [72, 11, 50, 54] learn compositional scene structure and estimate states through physical motions and visual de-animation. Recently, two papers propose pure physics-based methods [44, 39] that make heavy use of differentiable physics simulators, but they typically assume concepts in the scene are given as input. Our work is unique in that we learn video and language concepts from raw videos and infer dynamics information from a differentiable simulator, combining the benefits of both learning and physics.

## 3 Method

By integrating differentiable physics into the dynamic reasoning framework, VRDP jointly learns visual perception, language concepts, and physical properties of objects. The first two provide prior knowledge for optimizing the third one to reason about the physical world, and the optimized physical properties in turn help to learn better concepts. In the following, we first give an overview of our framework and then describe each of its components in detail.

### 3.1 Framework Overview

An overview of VRDP is illustrated in Fig. 1. It contains three components: a visual perception module, a concept learner, and a physics model. The input to the framework is a video and reasoning questions, where the former is processed by the visual perception module to get object trajectories and corresponding visual features, and the latter is parsed into executable symbolic programs with language concepts by the concept learner. Similar to DCL [58, 16], the concept learner first grounds object properties (*e.g.*, color and shape) and event concepts (*e.g.*, collision and moving) by aligning the

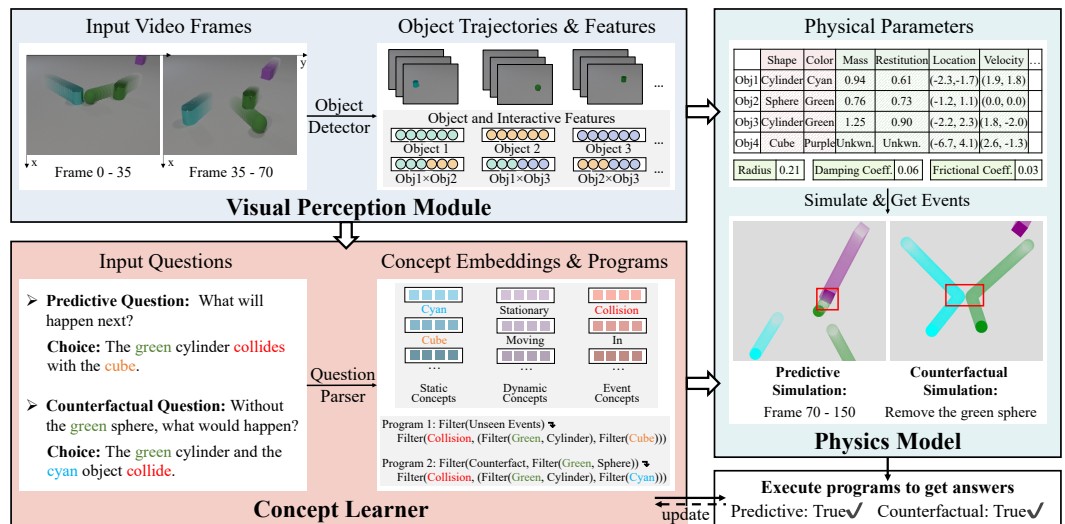

Figure 1: VRDP contains three components including a visual perception module, a concept learner, and a physics model. The perception module first runs an object detector [31] on individual frames to generate object proposals and connect them into trajectories based on motion heuristic. Then, the concept learner learns object- and event-based concepts, such as 'shape', 'moving', and 'collision', as prior knowledge for the physics model. Based on the obtained object trajectories and concepts, the differentiable physics engine estimates all dynamic and physical properties (*e.g.*, velocity $v$, angular velocity $\alpha$, restitution $r$, mass $m$, and coefficients of resistance $\lambda$) by comparing the simulated trajectories with the video observations. With these explicit physical parameters, the physic engine reruns the simulation to reason about future motion and causal events, which are then executed by a symbolic executor to get the answer. Stroboscopic imaging is applied for motion visualization.

visual features and the corresponding concept embeddings in the (explanatory or descriptive) program that does not require dynamic predictions, *e.g.*, "what is the shape of ...". With those perceptually grounded object trajectories and properties, the physical model then performs differentiable simulation to learn all physical parameters of the scene and objects by comparing the simulated trajectories with the video observations. After that, the physics engine simulates unseen trajectories for predictive and counterfactual scenarios and generates their features, in turn enabling the concept learner to finetune event concepts from the program that requires dynamic predictions, *e.g.*, "what will happen ..." and "what if ...". Finally, a symbolic executor executes the parsed programs with the dynamic predictions to get the answer.

## 3.2   Model Details

**Visual Perception Module**   Given a video with the number of frames $T$, the visual perception module parses the video frame-by-frame and associate the parsed objects in each frame into object trajectories $L = \{l^n\}_{n=1}^N$, where $l^n$ denotes the object trajectory of the $n^{th}$ object and $N$ is the number of the objects in the video. Specifically, we leverage a pretrained Faster R-CNN [31] as the object detector to get the Region of Interest (ROI) feature $f_t \in \mathbb{R}^{N \times D}$ and the object location of objects $b_t = [x_t^{2D}, y_t^{2D}, x_t^{BEV}, y_t^{BEV}] \in \mathbb{R}^{N \times 4}$ at frame $t$, where $D$ is the feature dimension, $(x_t^{2D}, y_t^{2D})$ denotes the normalized object bounding box center in the image coordinate frame, and $(x_t^{BEV}, y_t^{BEV})$ denotes the projected bird's-eye view (BEV) location in the BEV coordinate frame using the calibrated camera matrix. Following works [16, 28], we associate object proposals in adjacent frames by thresholding their intersection over union (IoU) and obtain the object trajectory $l^n = \{b_t^n\}_{t=1}^T$ for the $n^{th}$ object.

The visual perception module then constructs object and interactive representations for concept learning. The object representation $F_{\text{obj}} \in \mathbb{R}^{N \times (D+4T)}$ contains both appearance-based $F_a = \text{avg}(\{f_t\}_{t=1}^T)$ and trajectory-based feature $F_l = \{b_t\}_{t=1}^T$ for modeling static properties and dynamic concepts, respectively, where $\text{avg}(\cdot)$ here represents the average ROI feature over time. The interactive feature $F_{\text{pair}} \in \mathbb{R}^{T \times N \times N \times 12S}$, where $S$ denotes a fixed temporal window size, is built on every

pair of objects. It contains object trajectories $\{b_t^i\}_{t_0-S/2}^{t_0+S/2}, \{b_t^j\}_{t_0-S/2}^{t_0+S/2}$ of the objects $i$ and $j$ and their distance $\{\mathrm{abs}(b_t^i - b_t^j)\}_{t_0-S/2}^{t_0+S/2}$ to model the collision event of the objects at a specific moment $t_0$.

**Concept Learner** The concept learner grounds the physical and event concepts (*e.g.*, shape and color) as prior knowledge for the physics model from the video representation and language. It first leverages a question parser to translate the input questions and choices into executable neuro-symbolic programs where each language concept in the program is represented by a concept embedding. Similar to [58, 16], this work adopts a seq2seq model [6] with an attention mechanism to translate language into a set of symbolic programs, *e.g.*, retrieving objects with certain colors, getting future or counterfactual events, finding the causes of an event, thus decomposing complex questions into step-by-step dynamic visual reasoning processes. The concept learner assigns each concept in the program (*e.g.*, color, shape, and collision) a randomly initialized embedding $e \in \mathbb{R}^C$ so that the symbolic program can be formulated as differentiable vector operations.

After that, it projects the visual representation into concept embedding spaces and performs Nearest Neighbor (NN) search to quantize concepts for the object attributes and events. We implement the projection through a linear layer $\mathcal{P}(\cdot)$ and calculate the cosine similarity between two vectors in the embedding space for NN search. For example, the confidence score of whether the $n^{th}$ object is a cube can be represented by $[\cos(\mathcal{P}(F_a^n), e_{\mathrm{cube}}) - \mu]/\sigma$, where $e_{\mathrm{cube}}$ is the embedding of concept 'cube', $\mu$ and $\sigma$ are the shifting and scaling scalars, and $\mathcal{P}(\cdot)$ maps a $D$-dimensional visual feature into a $C$-dimensional vector in this case.

**Physics Model** The differentiable physics model captures objects' intrinsic physical properties and makes accurate dynamic predictions for reasoning. With the perceptually grounded object shapes and trajectories from the above two components of VRDP, it performs differentiable simulation to optimize the physical parameters of the scene and objects by comparing the simulation with the video observations $L$. Our physics model is implemented as an impulse-based differentiable rigid-body simulator [37, 62, 14]. It iteratively simulates a small time step of $\Delta t$ based on the objects' state in the BEV coordinate through inferring collision events, forces (including resistance and collision force), and impulses acting on the object, and updating the state of each object.

When an object moves on the ground with velocity $\overrightarrow{v}$ and angular velocity $\omega$, we consider three kinds of forces that affect the movement of the object: sliding friction, rolling resistance, and air resistance. We use $\lambda_1, \lambda_2, \lambda_3$ to denote their coefficients and have:

$$\overrightarrow{a} = \begin{cases} -\frac{\overrightarrow{v}}{|\overrightarrow{v}|}(\lambda_1 \mathrm{g} + \lambda_3|\overrightarrow{v}|^2) & \text{if the shape is not sphere} \\ -\frac{\overrightarrow{v}}{|\overrightarrow{v}|}(\lambda_2 \mathrm{g} + \lambda_3|\overrightarrow{v}|^2) & \text{if the shape is sphere} \end{cases} \qquad (1)$$

where $\mathrm{g} = 9.81\mathrm{m/s}^2$ is the standard gravity and $\overrightarrow{a}$ denotes the acceleration of the object, whose direction is opposite to the velocity. The velocity $\overrightarrow{v}$ and the location $\overrightarrow{l} = (x', y')$ are then updated accordingly by the second order Runge-Kutta (RK2) algorithm [13]. Similarly, the angular velocity $\omega$ also decreases at each time step due to the angular drag, and the angle $\alpha$ of the object is updated by the RK2 algorithm.

The physics engine checks whether the boundaries of two objects with radius $R$ are overlapped in the BEV coordinate frame to detect collision events. Based on the fact that the total momentum of an isolated system should be constant in the absence of net external forces, we compute the impulse of collided objects and ignore the friction caused by the collision. Let $(m_1, m_2)$, $(r_1, r_2)$, $(\alpha_1, \alpha_2)$, $(\overrightarrow{v_1}, \overrightarrow{v_2})$, $(\overrightarrow{l_1'}, \overrightarrow{l_2'})$ denote the mass, restitution, angle, velocity and BEV location of two collided objects at the moment of the collision, respectively; $\overrightarrow{d_1}, \overrightarrow{d_2}$ represent their collision unit directions that the force is acting on, where $\overrightarrow{d_1} + \overrightarrow{d_2} = \overrightarrow{0}$. The change of velocity $\Delta\overrightarrow{v_1}, \Delta\overrightarrow{v_2}$ at the moment of collision can be obtained by calculating the impulse on the collision direction:

$$\overrightarrow{\Delta v_1} = -(1 + r_1 r_2)(m_2/(m_1 + m_2))(\overrightarrow{d_1} \cdot (\overrightarrow{v_1} - \overrightarrow{v_2}))\overrightarrow{d_1}$$
$$\overrightarrow{\Delta v_2} = -(1 + r_1 r_2)(m_1/(m_1 + m_2))(\overrightarrow{d_2} \cdot (\overrightarrow{v_2} - \overrightarrow{v_1}))\overrightarrow{d_2}, \qquad (2)$$

the velocity $\overrightarrow{v}$ is then updated by $\overrightarrow{v} \leftarrow \overrightarrow{v} + \Delta\overrightarrow{v}$. Similarly, the angular velocity $\omega$ can be updated by $\omega \leftarrow \omega + \Delta\omega$, where $\Delta\omega$ is computed based on conservation of angular momentum.

Given an initial state of the scene and objects, our physics engine simulates force, impulse, and collision events and iteratively updates the state of each element. All physical parameters including

$R, \lambda, m, r, \alpha, \overrightarrow{v}, \overrightarrow{l}$ are initialized and then optimized with L-BFGS algorithm [56] by fitting the simulated trajectories $L' = \{(x'_t, y'_t)\}_{t=1}^T$ into the perceptual trajectories $L^{\text{BEV}} = \{(x_t^{\text{BEV}}, y_t^{\text{BEV}})\}_{t=1}^T$. To alleviate the difficulty of the optimization, we mark the time frame of each object's first collision by calculating the BEV distance between every pair and decompose the differentiable physical optimization and simulation into the following steps: 1) Since radius $R$ and resistance coefficients $\lambda$ are consistent in all videos, we use $K$ videos to jointly learn those physical parameters and fix them for the optimization of other sample-dependent parameters. 2) For each video, we then use the frames before the collision to optimize the collision-independent physical parameters, such as initial velocity $\overrightarrow{v_0}$, initial location $\overrightarrow{l_0}$, and initial angle $\alpha_0$. 3) With the above parameters learned and fixed, we optimize the remaining collision-dependent parameters, including mass $m$ and resistance $r$ of each object. This process follows the curriculum learning paradigm [12] by optimizing from fewer to more frames, *e.g.*, multi-step optimization on [0, 40], [0, 80], and [0, 128] frames, where the parameters in each step are initialized from the optimization of the previous step. 4) With all parameters of the physical model learned, the engine runs simulations and re-calculates the trajectory-based representations $F_l$ for answering counterfactual, descriptive, and explanatory questions. 5) For the predictive case, we leverage the learned physical model as initialization and re-optimize all sample-dependent parameters with only the last 20 frames to reduce the cumulative error over time.

**Symbolic Execution** As in [58, 16], we perform reasoning with a program executor, which is a collection of deterministic functional modules designed to realize all logic operations specified in symbolic programs. Its input consists of the parsed programs, learned concept embeddings, and visual representations, including the appearance-based feature $F_a$ from the visual perception module and the updated trajectory feature $F_l$ from the physics engine. Given a set of parsed programs, the program executor runs them step-by-step and derives the answer based on these representations. For example, the 'counting' program outputs the number of objects which meet specific conditions (*e.g.*, red sphere). In this process, the executor leverages the concept learner to filter out eligible objects.

Our reasoning process is designed fully differentiable w.r.t. the visual representations and the concept embeddings by representing all object states, events, and results of all operators in a probabilistic manner during training, supporting gradient-based optimization. Moreover, it works seamlessly with our explicit physics engine, which simulates dynamic predictions through real physical parameters, forming a symbolic and deterministic physical reasoning process. The whole reasoning process is fully transparent and step-by-step interpretable.

### 3.3 Training Objectives

Similar to [16, 79], we train the program parser with program labels using cross-entropy loss,

$$\mathcal{L}_{program} = -\sum_{j=1}^{J} \mathbf{1}\{y_p = j\} \log(p_j), \tag{3}$$

where $J$ is the size of the pre-defined program set, $p_j$ is the probability for the $j$-th program and $y_p$ is the ground-truth program label.

We optimize the physical parameters in the physical model by comparing the simulation trajectories with the video observations. All physical parameters including $R, \lambda, m, r, \alpha, \overrightarrow{v}, \overrightarrow{l}$ are initialized and then optimized with L-BFGS algorithm [56] by fitting the simulated trajectories $L' = \{(x'_t, y'_t)\}_{t=1}^T$ into the perceptual trajectories $L^{\text{BEV}} = \{(x_t^{\text{BEV}}, y_t^{\text{BEV}})\}_{t=1}^T$. We have:

$$\mathcal{L}_{Physics} = \|L' - L^{\text{BEV}}\|_2^2, \tag{4}$$

We optimize the feature extractor and the concept embeddings in the concept learner by question answering. We treat each option of a multiple-choice question as an independent boolean question during training. Specifically, we use cross-entropy loss to supervise open-ended questions and use mean square error loss to supervise counting questions. Formally, for counting questions, we have

$$\mathcal{L}_{QA,count} = (y_a - z)^2, \tag{5}$$

where $z$ is the predicted number and $y_a$ is the ground-truth number label. For other open-ended questions and multiple-choice questions, we have

$$\mathcal{L}_{QA,others} = -\sum_{a=1}^{A} \mathbf{1}\{y_a = a\} \log(p_a), \tag{6}$$

where $A$ is the size of the pre-defined answer set, $p_a$ is the probability for the $a$-th answer and $y_a$ is the ground-truth answer label.

## 4 Experiments

By recovering physics models of objects and their interactions from video and language, VRDP enjoys the following benefits: 1) high accuracy and full transparency, 2) superior data efficiency, and 3) high generalizability. In this section, we first evaluate the accuracy and data efficiency of VRDP on the widely used dynamic visual reasoning benchmark CLEVRER [79] and its subsets; we then validate the model's generalizability on adapting to new concepts with few-shot data; lastly, we experiment on the real-world dataset Real-Billiard [63] to show that VRDP works well in real-world dynamic prediction and reasoning.

**Datasets and Evaluation Settings** To validate the effectiveness of our method for reasoning about the physical world, we conduct main experiments on the CLEVRER [79] dataset, as it contains both language and physics cues such as rigid body collisions and dynamics, compared to other benchmarks that focus on either action understanding without physical inferring [49, 42] or temporal reasoning without language concepts [27, 8]. CLEVRER includes four types of question answering (QA): descriptive, explanatory, predictive, and counterfactual, where the first two types concern more on video understanding, while the latter two types involve physical dynamics and predictions in reasoning. Therefore, we mainly focus on the predictive and counterfactual questions in this work and use QA accuracy as the evaluation metric. Note the multi-choice question (explanatory, predictive, and counterfactual) contains multiple options. Only if all the options (per opt.) are answered correctly can it be regarded as a correct question (per ques.).

We then collect a few-shot physical reasoning dataset with novel language and physical concepts (*e.g.*, "heavier" and "lighter"), termed generalized CLEVRER, containing 100 videos (split into 25/25/50 for train/validation/test) with 375 options in 158 counterfactual questions. This dataset is supplementary to CLEVRER [79] for generalizing to new concepts with very few samples. For real-world scenarios, we conduct experiments on the Real-Billiard [63] dataset, which contains three-cushion billiards videos captured in real games for dynamics prediction. We generate 6 reasoning questions (*e.g.*, "will one billiard collide with ...?") for each video and evaluate both the prediction error and QA accuracy.

**Implementation Details** We follow the experimental setting in [79, 16] using a pre-trained Faster R-CNN model [31] to generate object proposals for each frame and training the language program parser with 1,000 programs for all question types. We implement three versions of VRDP models, where our unsupervised VRDP leverage a Slot-Attention model [57] to parse the objects unsupervisedly, while VRDP † use Faster-RCNN [31] as the object detector. In addition to our standard model that grounds object properties from question-answer pairs, we also train a variant (VRDP †‡) on CLEVRER with an explicit rule-based program executor [79] and object attribute supervision. The camera matrix is optimized from 20 training videos. We set $\Delta t = 0.004s$, $K = 10, S = 10$, and $T = 128$ for CLEVRER [79] and $T = 20$ for Real-Billiard [63]. More details of the dataset and settings can be found in Supplemental Materials.

### 4.1 Comparative Results on CLEVRER

We conduct experiments on CLEVRER against several counterparts: TVQA+ [49], Memory [22], IEP (V) [45], TbD-net (V) [59], HCRN [48], MAC [40], NS-DR [79], DCL [16], and Object-based Attention [19]. Among them, NS-DR [79] and DCL [16] are high-performance interpretable symbolic models, while Object-based Attention [19] is the state-of-the-art end-to-end method.

From Tab. 1 we observe that: 1) Counterfactual and predictive questions are more difficult than descriptive and explanatory ones as they require accurate physical dynamics and prediction hence our main focus. By reconstructing the physical world explicitly, our method outperforms all existing works on these two types by large margins. For example, VRDP † improves the per question accuracy of counterfactual questions by 11.5% and 79.7% compared to the best end-to-end [19] and neural-symbolic [16] counterparts.

2) The end-to-end model [19] improves the accuracy at the cost of losing model transparency and interpretability. However, by leveraging object attribute supervision and explicit program executors [79], our VRDP †‡ achieves new state-of-the-art overall performance on CLEVRER. It

| Methods | Overall | | Predictive | | Counterfactual | | Descriptive | Explanatory | |
|---|---|---|---|---|---|---|---|---|---|
| | per task | per ques. | per opt. | per ques. | per opt. | per ques. | | per opt. | per ques. |
| TVQA+ [49] | 37.2 | 57.3 | 70.3 | 48.9 | 53.9 | 4.1 | 72.0 | 63.3 | 23.7 |
| Memory [22] | 27.2 | 43.3 | 50.0 | 33.1 | 54.2 | 7.0 | 54.7 | 53.7 | 13.9 |
| IEP (V) [45] | 20.2 | 40.5 | 50.0 | 9.7 | 53.4 | 3.8 | 52.8 | 52.6 | 14.5 |
| TbD-net (V) [59] | 23.6 | 58.6 | 50.3 | 6.5 | 56.1 | 4.4 | 79.5 | 61.6 | 3.8 |
| HCRN [48] | 27.3 | 44.8 | 54.1 | 21.0 | 57.1 | 11.5 | 55.7 | 63.3 | 21.0 |
| MAC (V) [40] | 32.1 | 65.5 | 51.0 | 16.5 | 54.6 | 13.7 | 85.6 | 59.5 | 12.5 |
| MAC (V+) [40] [†] | 44.2 | 69.8 | 59.7 | 42.9 | 63.5 | 25.1 | 86.4 | 70.5 | 22.3 |
| NS-DR [79] [†‡] | 69.7 | 80.7 | 82.9 | 68.7 | 74.1 | 42.2 | 88.1 | 87.6 | 79.6 |
| NS-DR (NE) [79] [†‡] | 64.1 | 77.7 | 75.4 | 54.1 | 76.1 | 42.0 | 85.8 | 85.9 | 74.3 |
| DCL [16] [†] | 75.5 | 84.1 | 90.5 | 82.0 | 80.4 | 46.5 | 90.7 | 89.6 | 82.8 |
| DCL-Oracle [16] [†‡] | 75.6 | 84.5 | 90.6 | 82.1 | 80.7 | 46.9 | 91.4 | 89.8 | 82.0 |
| Object-based Attention [19] | 88.3 | 91.7 | 93.5 | 87.5 | 91.4 | 75.6 | 94.0 | 98.5 | 96.0 |
| VRDP (ours) | 82.9 | 86.9 | 91.7 | 83.8 | 89.9 | **75.7** | 89.8 | 89.1 | 82.4 |
| VRDP (ours) [†] | 86.6 | 89.4 | **94.5** | **89.2** | 92.5 | **80.7** | 91.5 | 90.9 | 85.2 |
| VRDP (ours) [†‡] | **90.3** | **92.0** | **95.7** | **91.4** | 94.8 | **84.3** | 93.4 | 96.3 | 91.9 |

Table 1: Question-answering accuracy of visual reasoning models on CLEVRER [79]. We report per-task and per-question overall accuracies, as well as per-option and per-question accuracies for each sub-task. Note that predictive and counterfactual questions that require dynamics and physical prediction are our focus. [†] denotes the method uses a supervised object detector, such as Faster/Mask R-CNN [31]. [‡] indicates the use of object properties (*i.e.*, shape, color, and material) as supervision.

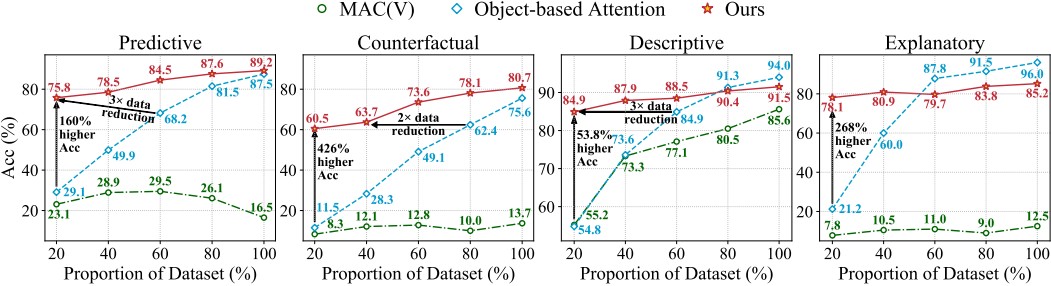

Figure 2: Comparisons of the data efficiency evaluation on four types of questions with MAC (V) [40] and Object-based Attention [19] trained with different proportion of the CLEVRER [79] dataset. Our method is highly data-efficient in that it achieves comparable results with the state-of-the-art counterpart [19] with 3× fewer data. It improves the reasoning accuracy significantly when fewer data (*e.g.*, 20%) are used.

closes the performance gap between interpretable models and state-of-the-art end-to-end methods. Moreover, it shows the flexibility of our physics model that can be combined with various physical concepts and program executors while achieving impressive performance.

3) We conducted ablative experiments to study the impact of pre-trained object detection modules of our framework by replacing the supervised visual model [31] in VRDP † with an unsupervised one [57] in VRDP. We observe that although the use of unsupervised detectors decreases the performance slightly, our framework still enjoys higher performance than previous methods in counterfactual and predictive questions.

4) Neither the neuro-symbolic nor end-to-end works employ explicit dynamic models with physical meanings. In contrast, our model is fully transparent with step-by-step interpretable programs and meaningful physical parameters powered by a differentiable engine.

## 4.2 Detailed Analysis

**Evaluation of Data Efficiency**   We evaluated the data efficiency of VRDP with two representative models: MAC (V) [40] and Object-based Attention [19]. From Fig. 2 we see that: VRDP is highly data-efficient. When the amount of data is reduced, the accuracy of our model drops slightly, while the performance of MAC (V) [40] and Object-based Attention [19] drops drastically due to insufficient data. For example, we improve the counterfactual accuracy of Object-based Attention [19] by 426%

**Question**: If the blue sphere were much **heavier**, which of the events that happened would not have happened?

**Choice**: The blue cylinder collides with the cube.

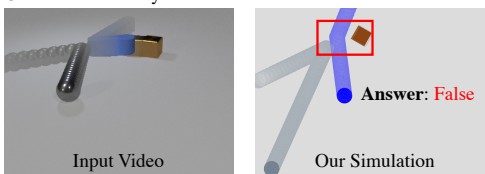

Input Video | Our Simulation

**Answer**: False

Figure 3: VRDP learns new concepts and accurately reasons about counterfactual events from few data on generalized CLEVRER.

| Methods | Per opt. | Per ques. |
|---|---|---|
| MAC (V) [40] | 63.8 | 22.0 |
| Object-based Attention [19] | 59.5 | 26.7 |
| VRDP (Ours) | **88.1** | **75.6** |

Table 2: Comparative results of generalizability evaluation under the few-shot setting. All models are first pretrained on CLEVRER and then fine-tuned with only 25 videos for adapting to generalized CLEVRER. VRDP can learn new concepts quickly with few-shot data.

| Methods | Overall | | Predictive | | Counterfactual | | Descriptive | Explanatory | |
|---|---|---|---|---|---|---|---|---|---|
| | per task | per ques. | per opt. | per ques. | per opt. | per ques. | | per opt. | per ques. |
| Baseline | 72.6 | 81.6 | 85.1 | 72.4 | 77.6 | 49.6 | 87.8 | 88.0 | 80.6 |
| + Collision-independent First | 81.3 | 87.8 | 86.1 | 72.8 | 89.3 | 74.1 | 91.3 | 91.9 | 86.9 |
| + Curriculum Optimization | 85.6 | 90.2 | 87.6 | 76.5 | 94.8 | 84.3 | 92.2 | 93.3 | 89.2 |
| + Re-optimization for Prediction | **90.3** | **92.0** | **95.7** | **91.4** | 94.8 | 84.3 | 93.4 | 96.3 | 91.9 |

Table 3: Ablation study on the optimization of physical parameters on CLEVRER [79]. The reasoning accuracy for the four types of questions is continuously increased through a better learning process.

under the setting of 20% data. Notably, our model uses 20% of the dataset to achieve comparable performance to other works that use 80% of data. This is because the components of VRDP, *e.g.*, perception module and question parser, can be trained with a small amount of data. More importantly, our physics model is built based on an explicit physics engine, which can be optimized from the trajectory of a single video.

**Evaluation of Generalizability**   This part studies the generalization capabilities of VRDP against MAC (V) [40] and Object-based Attention [19] on the generalized CLEVRER dataset. Tab. 2 shows our model outperforms other works by a large margin (75.6 vs. 26.7) on per question accuracy, demonstrating our model can quickly learn new concepts from few examples by reconstructing the physics world. An example of generalization with few-shot data is shown in Fig. 3. Our model learns a novel concept "heavier" from only 25 videos and the corresponding question-answer pairs. The simulation is then run with 5 times the mass to answer the question correctly.

**Ablation Study on the Learning of Physics Models**   In this work, sample-independent physical parameters $(R, \lambda)$ are learned from multiple training videos. In contrast, the sample-dependent parameters, such as $m, r, \alpha, v, l$, can only be learned with a single video, leading to difficulties in optimization, especially when there are many collisions. This part studies the optimization of these sample-dependent parameters by making comparisons among the following four simplified learning processes on CLEVRER [79]: 1) Baseline – optimize all target parameters directly from all frames simultaneously. 2) Collision-independent First – first use the frames before the collision to optimize collision-independent parameters for each object, including initial velocity $\overrightarrow{v_0}$, initial location $\overrightarrow{l_0}$, and initial angle $\alpha_0$; then optimize mass $m$ and restitution $r$ from all video frames. 3) Curriculum Optimization – optimize $m$ and $r$ by performing multiple steps on $[0, 40]$, $[0, 80]$, and $[0, 128]$ frames, where each step is initialized from the optimization of the previous step. 4) Re-optimization for Prediction (Full model) – leverage the learned physical parameters as initialization and re-optimize all sample-dependent parameters with the last 20 frames to reduce the cumulative error over time.

Tab. 3 shows that the performance continuously increases when more optimization steps are used, demonstrating the contribution of each part. The "Collision-independent First" rule offers the greatest improvement, especially for counterfactual questions, as counterfactual simulations only rely on the initial state. "Curriculum Optimization" improves all types of questions, and "Re-optimization for Prediction" re-calculates the dynamics of the last 20 frames, thus mainly affect predictive questions.

**Failure Analysis**   VRDP learns the physics model from object trajectories in videos and language concepts in question-answer pairs. It is data-efficient and robust enough to work well when there exists inaccurate perception or incorrect concept learning in some video frames. However, we noticed

**Question:** Will the red billiard collide with the top side of the billiard table?

**Answer:** True

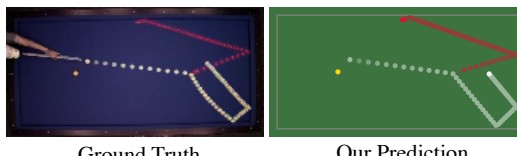

Ground Truth          Our Prediction

Figure 4: An example of physical simulation and question-answering on the real-world billiard dataset [63]. VRDP learns accurate physics parameters and infers the correct answer by simulation.

| Methods | S1 Err. ↓ | S2 Err. ↓ | QA Acc. (%)↑ |
|---|---|---|---|
| VIN [71] | 1.02 | 5.11 | 58.3 |
| OM [43] | 0.59 | 3.23 | 61.1 |
| CVP [77] | 3.57 | 6.63 | 58.3 |
| IN [10] | 0.37 | 2.72 | 69.4 |
| CIN [63] | 0.30 | 2.34 | 72.2 |
| **VRDP (Ours)** | **0.24** | **0.88** | **80.6** |

Table 4: Comparisons of the prediction error and question-answering accuracy on Real-Billiards. The rollout timesteps are chosen to be the same (S1) and twice (S2) as the training time ($T = 20$). The error is scaled by 1,000.

that the model might fail in the following cases: 1) If the object collides immediately after entering the image plane, there are insufficient frames before the collision to learn the initial velocity $v_0$. 2) If no collision occurs on an object, its restitution $r$ and mass $m$ cannot be optimized (unknown). We set default values for them. 3) The optimization becomes difficult if there are many cubes and collisions between them in the scene, because cube collisions (considering the sides and corners) are more complicated than sphere and cylinders'. These issues are challenging and will be our future work.

### 4.3 Comparative Results on Real-World Billiards

We also conduct experiments on the real-world dataset Real-billiard [63] with our supplemented question-answer pairs. Note that the billiard table is a chaotic system, and highly accurate long-term prediction is intractable. Fig. 4 shows an example of the ground truth video and our simulated prediction based on the perceptual grounded physics model. It can be seen that the predicted collision events and trajectories are of good quality. Tab. 4 evaluates the prediction errors under two different rollout timesteps and QA accuracy with 5 competitors: VIN [71], OM [43], CVP [77], IN [10], and CIN [63]. For the prediction task, the rollout timesteps are chosen to be the same (S1= $[0, T]$) and twice (S2= $[T, 2T]$) as the training time, where the training time $T = 20$. We refer interested readers to CIN [63] for more details. We find that VRDP is superior to these methods on both prediction and question answering tasks. Moreover, VRDP works well in long-term prediction. It reduces the S2 error on CIN [63] by 62.4%.

## 5 Conclusion

This work introduces VRDP, a unified framework that integrates powerful differentiable physics models into dynamic visual reasoning. It contains three mutually beneficial components: a visual perception module, a concept learner, and a differentiable physics engine. The visual perception module parses the input video into object trajectories and visual representations; the concept learner grounds language concepts and object attributes from question-answer pairs and the visual representations; with object trajectories and attributes as prior knowledge, the physics model optimizes all physical parameters of the scene and objects by differentiable simulation. With these explicit physical parameters, the physics model reruns the simulation to reason about future motion and causal events, which are then executed by a symbolic program executor to get the answer. Equipped with the powerful physics model, VRDP is of highly data-efficient and generalizable that adapts to novel concepts quickly with few-shot data. Moreover, both the explicit physics engine and the symbolic executor are step-by-step interpretable, making VRDP fully transparent. Extensive experiments on CLEVRER and Real-Billiards show that VRDP outperforms state-of-the-art dynamic reasoning methods by large margins.

## Acknowledgments and Disclosure of Funding

This work was supported by MIT-IBM Watson AI Lab and its member company Nexplore, ONR MURI, DARPA Machine Common Sense program, ONR (N00014-18-1-2847), and Mitsubishi Electric. Ping Luo was supported by the General Research Fund of Hong Kong No.27208720.

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
