# Supplementary Materials – Dynamic Visual Reasoning by Learning Differentiable Physics Models from Video and Language

**Mingyu Ding**
MIT CSAIL and HKU

**Zhenfang Chen**
MIT-IBM Watson AI Lab

**Tao Du**
MIT CSAIL

**Ping Luo**
HKU

**Joshua B. Tenenbaum**
MIT BCS, CBMM, CSAIL

**Chuang Gan**
MIT-IBM Watson AI Lab

## A    Appendix

In this section, we provide supplementary details of our VRDP [1]. First, we give more details of our physics model and the neuro-symbolic operations in the program executor. We then introduce the datasets we use and build, including a synthetic dataset (CLEVRER [10]), a real-world dataset (Real-Billiard [9]), and a newly built few-shot dataset (Generalized CLEVRER). After that, we detail the training settings and steps.

### A.1    Details of Physics Model

In this part, we provide supplementary details of our physics model. With the perceptually grounded object shapes and trajectories from the perception module and the concept learner of VRDP, our physics model performs differentiable simulation to optimize the physical parameters of the scene and objects by comparing the simulation $L'$ with the video observations $L^{\text{BEV}}$. The target bird's-eye view (BEV) trajectory $L^{\text{BEV}}$ is obtained by projecting the object center to the BEV coordinate. The Camera-to-BEV projection can be written as:

$$\begin{bmatrix} x \\ y \\ - \\ 1 \end{bmatrix}_{BEV} = \mathbf{K}^{-1} \cdot \begin{bmatrix} x \cdot z \\ y \cdot z \\ z \\ 1 \end{bmatrix}_{camera} \tag{1}$$

where $\mathbf{K}$ is the estimated camera matrix, $[x, y, z]_{camera}$ is the point in 2D image coordinates ($z_{camera}$ can be calculated from the camera matrix $\mathbf{K}$), $[x, y]_{BEV}$ denotes the horizontal position and vertical position of the projected point in BEV coordinates.

Based on the graphics programming language DiffTaichi [4], our physics model is implemented as an impulse-based differentiable rigid-body simulator. Based on conservation of momentum and angular momentum, it iteratively simulates a small time step of $\Delta t$ based on the objects' state in the BEV coordinate through inferring collision events, forces and impulses acting on the object, and updating the state of each object. In addition to calculating the acceleration based on the conservation of momentum in our main paper, we also calculate the angular acceleration based on the angular momentum. For example, we have: $\overrightarrow{M} = \overrightarrow{r} \times \overrightarrow{F}$ and $M = I\frac{\mathrm{d}\omega}{\mathrm{d}t}$, where $\overrightarrow{M}$ denotes moment of force, $\overrightarrow{F}$ is the applied force, and $\overrightarrow{r}$ is the distance from the applied force to object. The momentum of inertia $I$ is $1/6m(2R)^2$ for the cube, where $m$ represents its mass.

---

[1]Project page: http://vrdp.csail.mit.edu/

35th Conference on Neural Information Processing Systems (NeurIPS 2021).

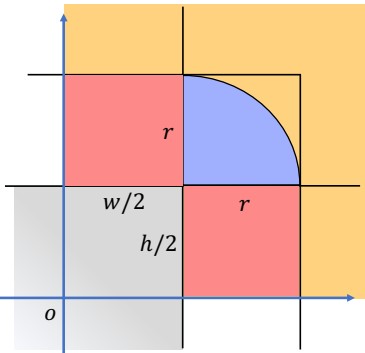

Figure 1: An illustration of circle-rectangle collision detection. The gray part denotes the rectangle (cube) and we transform the origin to the center of the rectangle so that the coordinate axis is parallel to its side. For each simulation step: we consider three situations: 1) if the center of the circle is in the orange area, the circle and the rectangle do not collide; 2) if the circle center is in the red area, the circle collides with the rectangle and the collision direction is perpendicular to the coordinate axis; 3) if the circle center falls in the purple area, the circle and the rectangle collide and the collision direction is perpendicular to the tangent of the collision position on the circle.

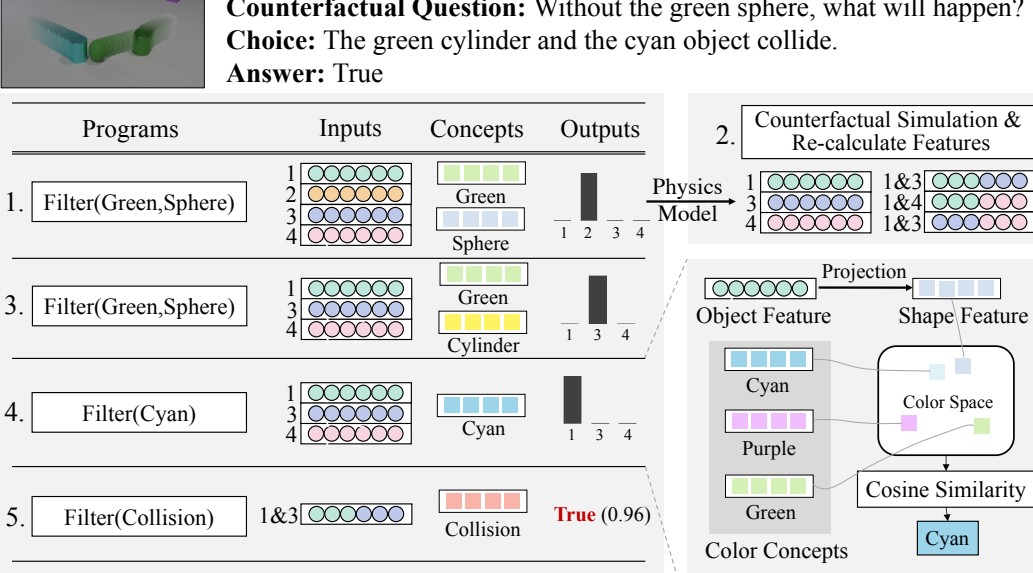

Figure 2: An illustration of the reasoning process of the program executor and concept learner. The program executor executes the parsed programs (*e.g.*, Filter_static_concept (color, shape, material)) step-by-step with the visual representations and language concepts. For each step, it leverages the concept learner or physical model to filter specific targets or simulate/predict new visual trajectories.

In this work, we perform collision detection between circles and rectangles in BEV view. Fig. 1 shows the illustration of our circle-rectangle collision detection algorithm. We project the center of the rectangle (the gray part) to the origin so that the coordinate axis is parallel to its side. Then the area outside the rectangle is divided into three parts that the center of the circle can fall: 1) collision with the sides of the square (red); 2) collision with the corners of the square (purple); 3) no collision (orange). The implementation of circle-circle and rectangle-rectangle collisions is similar.

## A.2 Details of Neuro-Symbolic Programs

Following DCL [1], we represent the objects, events and moments through learnable embeddings and quantize the static and dynamic concepts to perform temporal and causal reasoning. In this part, we list all the available data types and operations for CLEVRER in Tab. 1. We refer interested readers to DCL [1] for more details.

We also visualize the reasoning process of an example step-by-step in Fig. 2. It shows how we get the correct answer for the counterfactual question 'Without the green sphere, what will happen?' with a choice 'The green cylinder and the cyan object collide'. After the first program

'Filter_static_concept(all objects, green sphere)' is executed, the executor removes the retrieved object, reruns the simulation to get counterfactual trajectories, and updates the visual features. After that, the executor runs the remaining programs and gets the final answer 'True' with a probability of 0.96 calculated through the cosine distance in the concept learner.

## A.3  Details of Datasets

**CLEVRER**  CLEVRER [10] is a diagnostic video dataset for systematic evaluation of computational models on a wide range of reasoning tasks. Objects in CLEVRER videos adopt similar compositional intrinsic attributes as in CLEVR [5], including three shapes (cube, sphere, and cylinder), two materials (metal and rubber), and eight colours (gray, red, blue, green, brown, cyan, purple, and yellow). All objects have the same size, same friction coefficient (except the sphere that rolling on the ground), so no vertical bouncing occurs during the collision. Each object has a different mass and a different restitution coefficient. CLEVRER introduces three types of events: enter, exit and collision, each of which contains a fixed number of object participants: 2 for collision and 1 for enter and exit. The objects and events form an abstract representation of the video.

CLEVRER includes four types of question: descriptive (*e.g.*'what color'), explanatory ('what's responsible for'), predictive ('what will happen next'), and counterfactual ('what if'), where the first two types concern more on video understanding and temporal reasoning, while the latter two types involve physical dynamics and predictions in reasoning. Therefore, we mainly focus on the predictive and counterfactual questions in this work. CLEVRER consists of 2,000 videos, with a number of 1,000 training videos, 5,000 validation videos, and 5,000 test videos. It also contains 219,918 descriptive questions, 33,811 explanatory questions, 14,298 predictive questions, and 37,253 counterfactual questions. In this paper, we tune the model using the validation set and evaluate it with the test set.

**Generalized CLEVRER**  To evaluate the generalizability of reasoning methods, we collect a few-shot physical reasoning dataset with novel language and physical concepts (*e.g.*, 'heavier' and 'lighter'), termed generalized CLEVRER, containing 100 videos (split into 25/25/50 for train/validation/test) with 375 options in 158 counterfactual questions. This dataset is supplementary to CLEVRER [10] for generalizing to new concepts (*i.e.*, heavier, lighter) with very few samples. All videos last for 5 seconds and are generated by a physics engine [2] that simulates object motion plus a graphs engine that renders the frames. It has the same video settings (objects and events settings) with CLEVRER but different questions/concepts, *e.g.*, "What would happen if the blue sphere were heavier?", we generate the ground truth video in the counterfactual case by setting five times the weight and perform the physical simulation with Bullet [2]. In this work, we evaluate the QA accuracy of this dataset.

**Real-Billiard**  For real-world scenarios, we conduct experiments on the Real-Billiard [9] dataset, which contains three-cushion billiards videos captured in real games for dynamics prediction. There are 62 training videos with 18,306 frames, and 5 testing videos with 1,995 frames. The bounding box annotations are from an off-the-shelf ResNet-101 FPN detector [6] pretrained on COCO [7] and fine-tuned on a subset of 30 images from our dataset. Wrong detections are manually filtered out. We generate 6 reasoning questions (*e.g.*, "will one billiard collide with ...?") for each video and evaluate both the prediction error and QA accuracy.

## A.4  Details of Training Settings

As in [10, 1], we use a pre-trained Faster R-CNN model [3] that is trained on 4,000 video frames randomly sampled from the training set with object masks and attribute annotations to generate object proposals for each frame. We train the language program parser with 1,000 programs for all question types. All deep modules (concept learner and program executor) are trained using Adam optimizer for 40 epochs on 8 Nvidia 1080Ti GPUs and the learning rate is set to $10^{-4}$. The camera matrix is optimized from 20 training videos. We set $\Delta t = 0.004$s, $D = 256, C = 64, K = 10, S = 10$, and $T = 128$ for CLEVRER [10] and $T = 20$ for Real-Billiard [9]. In addition to our standard model that grounds object properties from question-answer pairs, we also train a variant (VRDP †) on CLEVRER with an explicit rule-based program executor [10] and object attribute supervisions (attribute annotation in 4000 frames learned by the Faster R-CNN model).

For the physical model, we use the L-BFGS optimizer [8] with an adaptive learning rate to optimize all physical parameters. The optimization terminates when it reaches a certain number of steps or the loss is less than a certain value. In all experiments, the number of the optimization step is set to 20. The loss threshold is set to 0.0005 for the learning of collision-independent parameters (*i.e.*, initial velocity, initial location, and initial angle), and 0.0002, 0.001, 0.01 for the optimization of collision-dependent parameters (mass and restitution) on [0, 40], [0, 80], and [0, 128] frames, respectively.

The training of VRDP can be summarized into three stages. First, we extract the visual features directly from the video by the visual perception module, and learn language concepts in the concept learner from all descriptive and explanatory questions; second, we optimize all physical parameters by using the perceived trajectories and the learned concepts; third, after obtaining the physics model, we re-calculate the visual features from the simulated trajectories and finetune language concept embeddings from all question types, including predictive and counterfactual questions. During this training process, the three parts of VRDP are integrated seamlessly and benefit each other.

### A.5 Visualizations

We show visualization examples (including failure cases) on CLEVRER [10] in Fig. 3 and Fig. 4. We also show examples on Real-Billiards [9] in Fig. 5. These figures show that our model can accurately learn physical parameters from video and language and perform causal simulations, predictive simulations, and counterfactual simulations for dynamic visual reasoning. Note that the billiard table is a chaotic system, and highly accurate long-term prediction is intractable. For more failure analysis, please refer to our main paper.

## Broader Impact

Our work focuses on dynamic visual reasoning about object interactions, dynamics, and physics with question answering, which is central to human intelligence and a key goal of artificial intelligence. We envision that the work will benefit a wide range of applications involving cognition and reasoning, such as robot control. The proposed method improves the accuracy, interpretability, and robustness of these applications, ultimately leading to better safety. We do not foresee obvious undesirable ethical/social impacts at this moment.

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

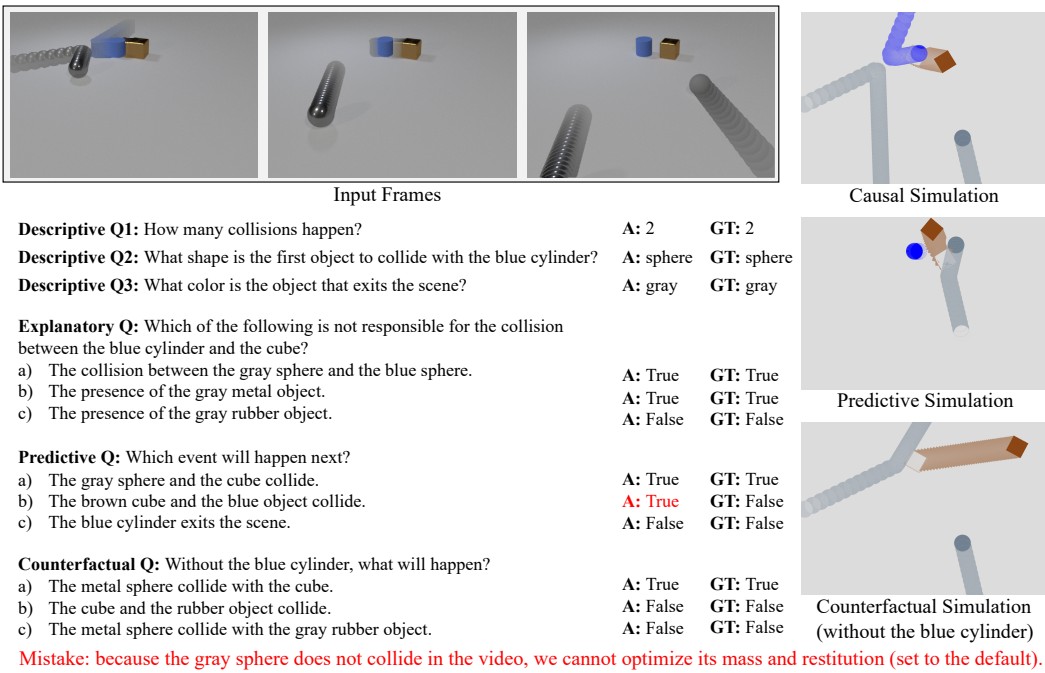

Input Frames — Causal Simulation

**Descriptive Q1:** How many collisions happen? — A: 2 — GT: 2

**Descriptive Q2:** What shape is the first object to collide with the blue cylinder? — A: sphere — GT: sphere

**Descriptive Q3:** What color is the object that exits the scene? — A: gray — GT: gray

**Explanatory Q:** Which of the following is not responsible for the collision between the blue cylinder and the cube?
a) The collision between the gray sphere and the blue sphere. — A: True — GT: True
b) The presence of the gray metal object. — A: True — GT: True
c) The presence of the gray rubber object. — A: False — GT: False

Predictive Simulation

**Predictive Q:** Which event will happen next?
a) The gray sphere and the cube collide. — A: True — GT: True
b) The brown cube and the blue object collide. — A: True — GT: False
c) The blue cylinder exits the scene. — A: False — GT: False

**Counterfactual Q:** Without the blue cylinder, what will happen?
a) The metal sphere collide with the cube. — A: True — GT: True
b) The cube and the rubber object collide. — A: False — GT: False
c) The metal sphere collide with the gray rubber object. — A: False — GT: False

Counterfactual Simulation (without the blue cylinder)

Mistake: because the gray sphere does not collide in the video, we cannot optimize its mass and restitution (set to the default).

---

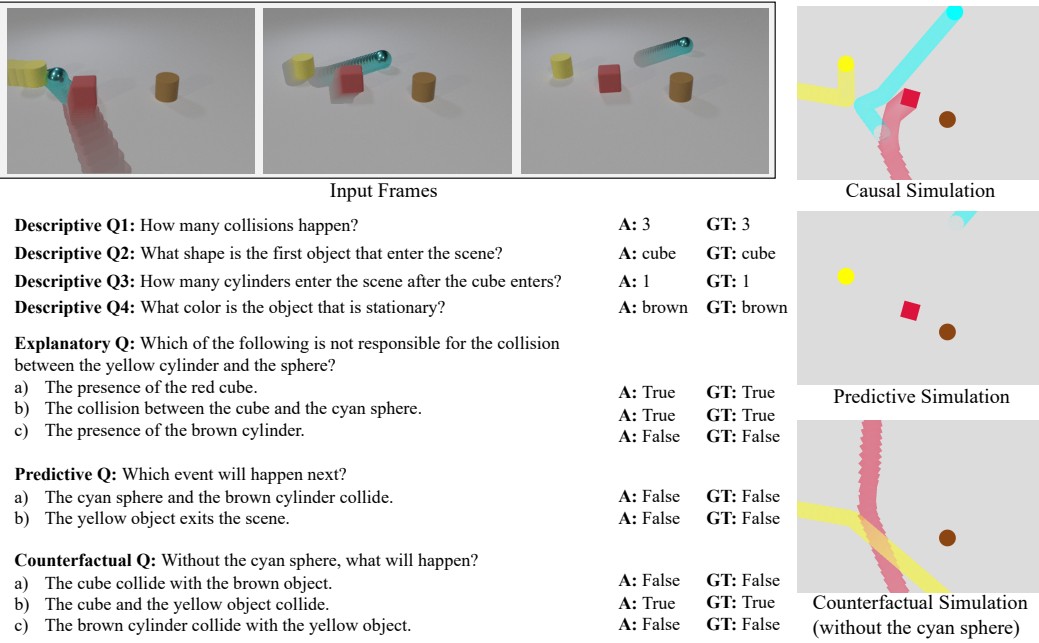

Input Frames — Causal Simulation

**Descriptive Q1:** How many collisions happen? — A: 3 — GT: 3

**Descriptive Q2:** What shape is the first object that enter the scene? — A: cube — GT: cube

**Descriptive Q3:** How many cylinders enter the scene after the cube enters? — A: 1 — GT: 1

**Descriptive Q4:** What color is the object that is stationary? — A: brown — GT: brown

**Explanatory Q:** Which of the following is not responsible for the collision between the yellow cylinder and the sphere?
a) The presence of the red cube. — A: True — GT: True
b) The collision between the cube and the cyan sphere. — A: True — GT: True
c) The presence of the brown cylinder. — A: False — GT: False

Predictive Simulation

**Predictive Q:** Which event will happen next?
a) The cyan sphere and the brown cylinder collide. — A: False — GT: False
b) The yellow object exits the scene. — A: False — GT: False

**Counterfactual Q:** Without the cyan sphere, what will happen?
a) The cube collide with the brown object. — A: False — GT: False
b) The cube and the yellow object collide. — A: True — GT: True
c) The brown cylinder collide with the yellow object. — A: False — GT: False

Counterfactual Simulation (without the cyan sphere)

Figure 3: Visualization (1) of the videos and question-answering results of our VRDP on CLEVRER. We highlighted our failure in red and explained the cause of it.

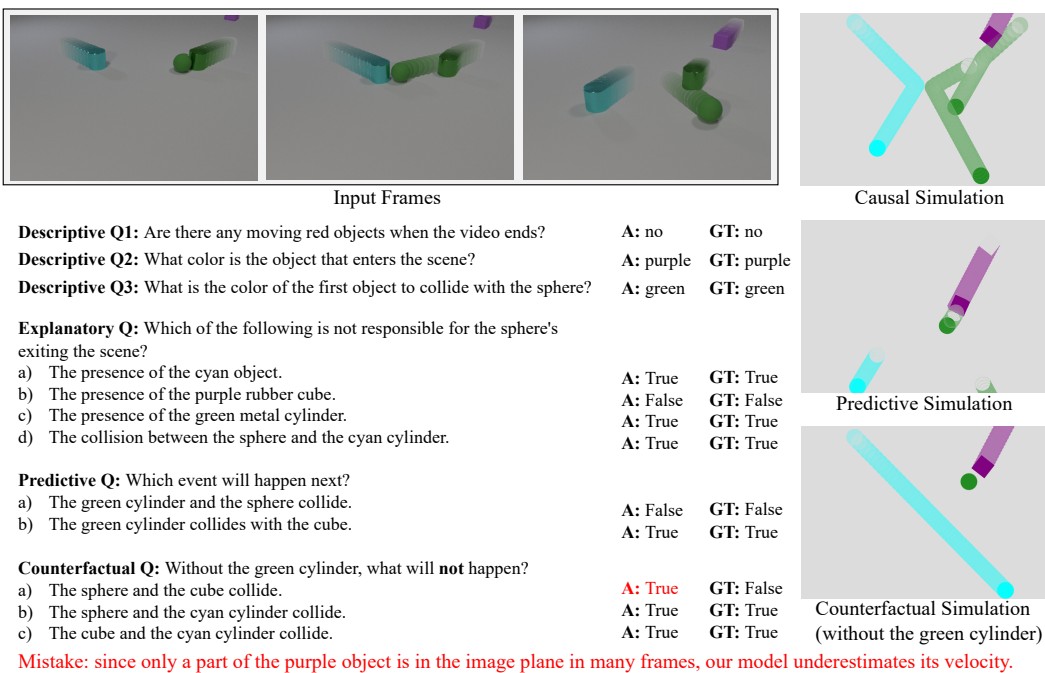

Input Frames | Causal Simulation

**Descriptive Q1:** Are there any moving red objects when the video ends?  **A:** no  **GT:** no

**Descriptive Q2:** What color is the object that enters the scene?  **A:** purple  **GT:** purple

**Descriptive Q3:** What is the color of the first object to collide with the sphere?  **A:** green  **GT:** green

**Explanatory Q:** Which of the following is not responsible for the sphere's exiting the scene?
a) The presence of the cyan object.  **A:** True  **GT:** True
b) The presence of the purple rubber cube.  **A:** False  **GT:** False
c) The presence of the green metal cylinder.  **A:** True  **GT:** True
d) The collision between the sphere and the cyan cylinder.  **A:** True  **GT:** True

**Predictive Q:** Which event will happen next?
a) The green cylinder and the sphere collide.  **A:** False  **GT:** False
b) The green cylinder collides with the cube.  **A:** True  **GT:** True

**Counterfactual Q:** Without the green cylinder, what will **not** happen?
a) The sphere and the cube collide.  **A:** True  **GT:** False
b) The sphere and the cyan cylinder collide.  **A:** True  **GT:** True
c) The cube and the cyan cylinder collide.  **A:** True  **GT:** True

Predictive Simulation

Counterfactual Simulation (without the green cylinder)

Mistake: since only a part of the purple object is in the image plane in many frames, our model underestimates its velocity.

---

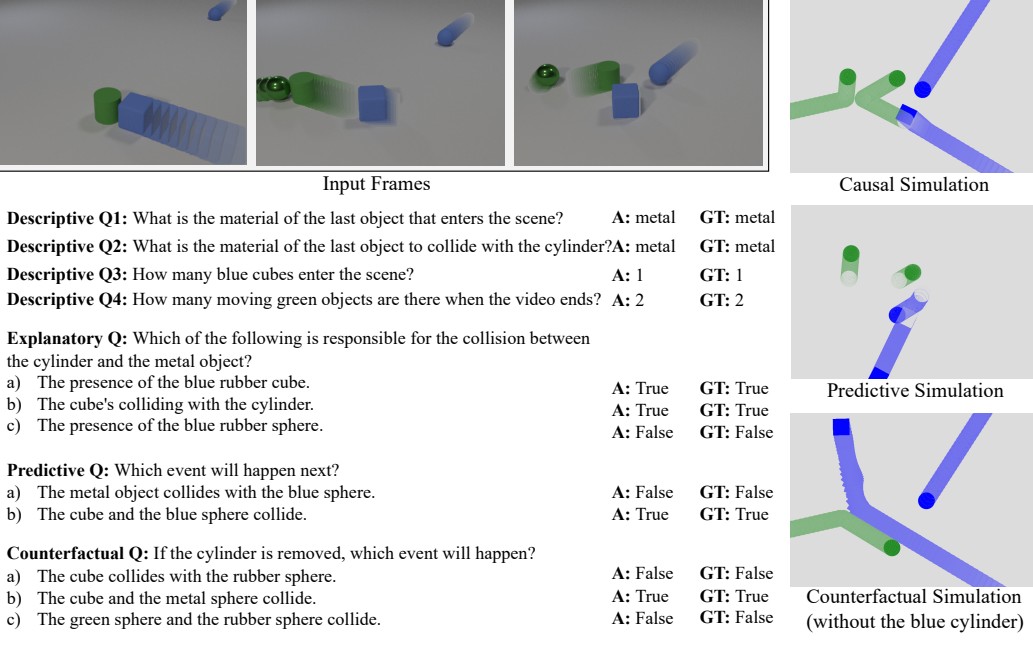

Input Frames | Causal Simulation

**Descriptive Q1:** What is the material of the last object that enters the scene?  **A:** metal  **GT:** metal

**Descriptive Q2:** What is the material of the last object to collide with the cylinder?  **A:** metal  **GT:** metal

**Descriptive Q3:** How many blue cubes enter the scene?  **A:** 1  **GT:** 1

**Descriptive Q4:** How many moving green objects are there when the video ends?  **A:** 2  **GT:** 2

**Explanatory Q:** Which of the following is responsible for the collision between the cylinder and the metal object?
a) The presence of the blue rubber cube.  **A:** True  **GT:** True
b) The cube's colliding with the cylinder.  **A:** True  **GT:** True
c) The presence of the blue rubber sphere.  **A:** False  **GT:** False

**Predictive Q:** Which event will happen next?
a) The metal object collides with the blue sphere.  **A:** False  **GT:** False
b) The cube and the blue sphere collide.  **A:** True  **GT:** True

**Counterfactual Q:** If the cylinder is removed, which event will happen?
a) The cube collides with the rubber sphere.  **A:** False  **GT:** False
b) The cube and the metal sphere collide.  **A:** True  **GT:** True
c) The green sphere and the rubber sphere collide.  **A:** False  **GT:** False

Predictive Simulation

Counterfactual Simulation (without the blue cylinder)

Figure 4: Visualization (2) of the videos and question-answering results of our VRDP on CLEVRER. We highlighted our failure in red and explained the cause of it.

| Type | Operation | Signature |
|---|---|---|
| Input Operations | `Start`
Returns the special "start" event | $() \rightarrow event$ |
| | `end`
Returns the special "end" event | $() \rightarrow event$ |
| | `Objects`
Returns all objects in the video | $() \rightarrow objects$ |
| | `Events`
Returns all events happening in the video | $() \rightarrow events$ |
| | `UnseenEvents`
Returns all future events happening in the video | $() \rightarrow events$ |
| Output Operations | `Query_color`
Returns the color of the input object | $(object) \rightarrow color$ |
| | `Query_material`
Returns the material of the input objects | $(object) \rightarrow material$ |
| | `Query_shape`
Returns the shape of the input objects | $(object) \rightarrow shape$ |
| | `Count`
Returns the number of the input objects/ events | $(objects) \rightarrow int$
$(events) \rightarrow int$ |
| | `Exist`
Returns "yes" if the input objects is not empty | $(objects) \rightarrow bool$ |
| | `Belong_to`
Returns "yes" if the input event belongs to the input event sets | $(event, events) \rightarrow bool$ |
| | `Negate`
Returns the negation of the input boolean | $(bool) \rightarrow bool$ |
| Physics Operations | `Counterfactual_simulation`
Perform simulation with the object removed | $(object) \rightarrow events, representations$ |
| | `Predictive_simulation`
Perform simulation after the video ends | $(objects) \rightarrow events, representations$ |
| | `Apply_heavier`
Assign the object five times its weight before the counterfactual simulation | $(object) \rightarrow object$ |
| | `Apply_lighter`
Assign the object one-fifth of its weight before the counterfactual simulation | $(object) \rightarrow object$ |
| Object Filter Operations | `Filter_static_concept`
Select objects from the input list with the input static concept | $(objects, concept) \rightarrow objects$ |
| | `Filter_dynamic_concept`
Selects objects in the input frame with the dynamic concept | $(objects, concept, frame) \rightarrow objects$ |
| | `Unique`
Return the only object in the input list | $(objects) \rightarrow object$ |
| Event Filter Operations | `Filter_in`
Select incoming events of the input objects | $(events, objects) \rightarrow events$ |
| | `Filter_out`
Select existing events of the input objects | $(events, objects) \rightarrow events$ |
| | `Filter_collision`
Select all collisions that involve an of the input objects | $(events, objects) \rightarrow events$ |
| | `Get_col_partner`
Return the collision partner of the input object | $(event, object) \rightarrow object$ |
| | `Filter_before`
Select all events before the target event | $(events, events) \rightarrow events$ |
| | `Filter_after`
Select all events after the target event | $(events, events) \rightarrow events$ |
| | `Filter_order`
Select the event at the specific time order | $(events, order) \rightarrow event$ |
| | `Filter_ancestor`
Select all ancestors of the input event in the causal graph | $(event, events) \rightarrow events$ |
| | `Get_frame`
Return the frame of the input event in the video | $(event) \rightarrow frame$ |
| | `Unique`
Return the only event in the input list | $(events) \rightarrow event$ |

Table 1: All neuro-symbolic operations on the CLEVRER dataset [10]. Our model contains five types of operations, including input, output, physics, object filter, and event filter operations. In this table, "order" denotes the chronological order of an event, *e.g.*"First", "Second" and "Last"; "static concept" denotes object-level static concepts like "Blue", "Cube" and "Metal"; "dynamic concept" represents object-level dynamic concepts like "Moving" and "Stationary"; and "representations" denotes the visual features that are calculated from object trajectories.

**Q1:** Will the yellow billiard collide with the right side of the billiard table?

**Ours:** True
**GT:** True

**Q2:** Will the yellow billiard collide with the top side of the billiard table?

**Ours:** False
**GT:** False

**Q3:** Will the white billiard collide with the right side of the billiard table?

**Ours:** True
**GT:** True

**Q4:** Will the white billiard collide with the top side of the billiard table?

**Ours:** True
**GT:** True

**Q1:** Will the yellow billiard collide with the right side of the billiard table?

**Ours:** True
**GT:** True

**Q2:** Will the yellow billiard collide with the red billiard?

**Ours:** False
**GT:** False

**Q3:** Will the yellow billiard collide with the bottom side of the billiard table?

**Ours:** False
**GT:** False

**Q4:** Will the yellow billiard collide with the white billiard?

**Ours:** False
**GT:** False

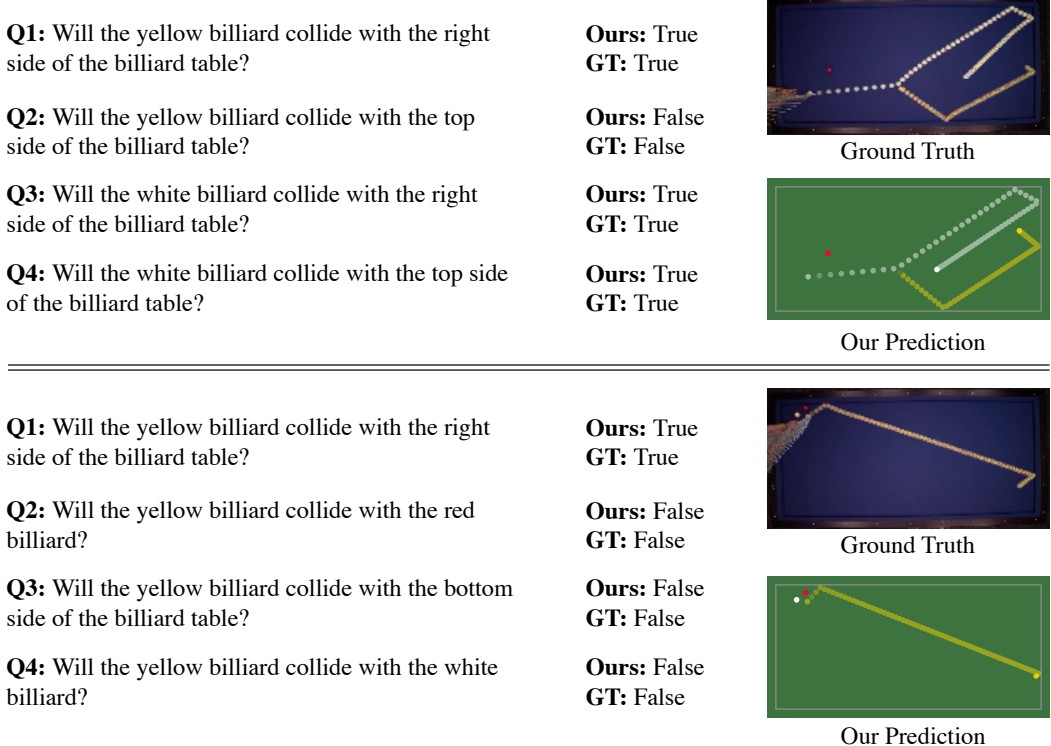

Figure 5: Visualization examples of the videos and question-answering results of our VRDP on Real-Billiards. Note that the billiard table is a chaotic system, and highly accurate long-term prediction is intractable.