# OpenReview forum: "Dynamic Visual Reasoning by Learning Differentiable Physics Models from Video and Language"
_NeurIPS.cc/2021/Conference — NeurIPS 2021 Poster_

### Official Review · Reviewer_BwFB · 2021-07-12

**Rating:** 6
**Confidence:** 3

**Summary:**

This paper introduces VRDP, a method that learns to answer predictive, counterfactual, explanatory and descriptive yes or no questions about videos depicting the evolution of physical scenes. VRDP is composed of three modules, an object detector, a neuro-symbolic concept learner and differentiable physics model.

The paper shows that VRDP surpasses or closely matches the SOTA on two datasets, a synthetic dataset CLEVERER and a dataset of real world videos.

**Ethical Concerns:**

None.

**Limitations And Societal Impact:**

Limitations and societal impact are properly addressed.

**Main Review:**

This paper introduces VRDP, a system that learns to perform question answering on videos of physical scenes.

The paper is well written, well organized and it properly places the contributions presented in the context of existing and related work.

I like the idea of combining distinct modules for visual perception, concept learning and physics simulation into a question answering system. I am impressed with the results and I would love to make some suggestions, which I hope can addressed in a revised version, that I think might help make this a more impactful paper.

1) The manuscript claims that VRDP is more interpretable than alternative approaches, however this is never defined or quantified. In particular, there is a large body of work on interpreting transformer based systems and the original Object Based Attention paper itself has an extensive attention analysis. Could the authors clarify what is meant by interpretability and provide qualitative or quantitative comparisons with alternative approaches? These could include e.g. linear readouts from transformer embeddings.

2) VRDP is substantially more data efficient than Object Based Attention. However, VRDP uses a pre-trained object detection module, whereas Object Based Attention trains its system end to end. It would be good to maybe include a discussion of this, it's certainly a positive to not have to reinvent the wheel with every new experiment, but one could imagine situations where end to end training could be beneficial. One possible way to take a crack at this would be to re-train an Object Based Attention system based on the output of VRDP's visual perception module and see if the sample complexity gap is still there.

3) The comparison to the current state of the art is limited to a single dataset , could the two additional datasets used in Object Based Attention be included as well? If not, could the authors please provide a discussion of what makes the comparison hard or impossible?

I would like to thank the authors for sharing these cool ideas and for writing a clear and easy to follow paper.
Physics based question answering directly from perception is an open, interesting and relevant problem. The results presented here on the CLEVERER dataset are promising, and the interpretability and reduced sample complexity are exciting prospects for a system such as this.

I hope the authors agree that my suggestions might make the paper stronger and take them into consideration. In its current form my impression is that this paper is not ready to be shared for the wider community because the results section is limited and it does not substantiate claims of improved sample complexity and interpretability.

I think these could easily be addressed in a revised version.

Thank you again for sharing these cool ideas, and I look forward to your rebuttal. -- All the best.

**Time Spent Reviewing:**

2

---

> ### Author Response · Authors · 2021-08-10
> **Response to Reviewer #BwFB**
>
> Thank you very much for the constructive comments.
>
> **Q1. [Qualitative or quantitative analysis of interpretability.]** \
> Good question. Our reasoning process is **explicit, transparent, and step-by-step interpretable**, while the reasoning of Object-Based Attention [19] is **implicit and hard to diagnose**. For example, humans reason about future collision events by first observing the two objects (visual perception), then intuitively estimate their attributes and moving directions (concept learning and intuitive physics), and then predict whether they will collide (simulation). Our reasoning process is amenable to human interpretation. It is easy to diagnose each component when the reasoning result is wrong, e.g., perceived wrong trajectories or learned wrong attributes/physical parameters. In contrast, it is difficult for Object-Based Attention [19] to identify the reason for the failure as their reasoning process is implicit.
>
> Specifically, our model employs a modularized design that all intermediate outputs are interpretable and have clear meanings, such as observed trajectories, color concepts, and symbolic programs. We know the concept of each word and the logic behind the symbolic executor. Also, since explicit physical properties like mass, friction, and restitution are inferred in the model, the dynamic prediction process is interpretable. In this way, our entire reasoning process is **explicit and step-by-step interpretable**.
>
> In contrast, the input of Object-Based Attention [19] includes object-based deep features and question language embeddings, and the output is directly the answer. Though extensive attention analysis has been made for understanding the "black box" deep model, its interpretability is only on the attention/network level, not the reasoning framework level.
>
>
> **Q2. [The impact of pre-trained object detection modules on data efficiency of frameworks.]** \
> Good suggestion. The data efficiency of our framework is mainly achieved by our symbolic representation and differentiable physics engine that can learn physical parameters from a few or even a single video. For the visual perception part, both our framework and Object-Based Attention [19] leverage pretrained visual models (Faster RCNN vs. MONet [A]) for object-based feature extraction. The difference is that our Faster-RCNN is trained with supervision while their MONet is trained unsupervised.
>
> We did try to reimplement [19] and replace their MONet feature with the Faster RCNN feature, but we observed no gains. This may be because: (1) The number of object representations extracted by Faster RCNN varies according to the number of objects in an image. However, the number of slots in MONet [A] is fixed. Though zero-pad was used in implementation, the two types of features are mismatched. (2) MONet [A] learns by reconstruction of the scene, so its features contain more object attributes and location information, while the Faster RCNN features are only used for object detection.
>
> Also, we conducted ablative experiments to study the impact of pre-trained object detection modules on the data efficiency of our framework by replacing the supervised visual model with an unsupervised one. The results are summarized in the following table.
>
> | Method (Visual Model)              |       Data Proportion            | Overall | Predictive | Counterfactual | Descriptive | Explanatory     |
> | :--------------------------------------------- | :-----: | :--------: | :------------: | :---------: | :---------: | :--: |
> | **[19]** (Unsupervised Detector) |       20%        | 29.2 |    29.1    |      11.5      |    54.8     |    21.2     |
> | **Ours** (Unsupervised Detector) |       20%        | 67.8 | 70.8 | 50.2 |    81.8     |    68.6     |
> | **Ours** (Supervised Detector) | 20% | 74.8 | 75.8 | 60.5 | 84.9 | 78.1 |
> |     |    |     |     |     |     |     |
> | **[19]** (Unsupervised Detector) | 40% |  53.0  |    49.9    |      28.3      |    73.6     |    60.0     |
> | **Ours** (Unsupervised Detector) | 40% |  70.6   |    72.3    |      52.3      |    84.0     |    73.7     |
> | **Ours** (Supervised Detector) | 40% |  77.8   |    78.5    |      63.7      |    87.9     |    80.9     |
>
> From the table, we observe that although the use of unsupervised detectors decreases the performance slightly, our framework still enjoys higher data efficiency than [19]. With the same unsupervised visual feature, our model outperforms [19] by a large margin (e.g., 67.8% vs. 29.2% using 20% data) under the low data regime.
>
> [A] Burgess, Christopher P., et al. "Monet: Unsupervised scene decomposition and representation." arXiv preprint arXiv:1901.11390.
>
>
> **Q3. [The comparison is limited to a single dataset.]** \
> We respectfully push back the comment that "the comparison to the current state of the art is limited to a single dataset.". In addition to the CLEVRER dataset, we also conducted the experiments on a real-world billiards video dataset [59] (Section 4.3) and a new generalized few-shot physics reasoning dataset (Section 4.2, generalizability). We believe these results could further verify the effectiveness and generalization of our work.
>
> We would also like to reiterate that our work aims to learn differentiable physics models from video and language. The mentioned two datasets, CATER [23] and ACRE [B], are of different research sub-directions with CLEVRER. They have neither **physical cues** nor **question-answer pairs**. Thus the two datasets are unsuitable for evaluating inferring physics models and learned concepts from video and language (see our related work in lines 89-93).
>
> [B] Chi Zhang, Baoxiong Jia, Mark Edmonds, Song-Chun Zhu, and Yixin Zhu. Acre: Abstract causal reasoning beyond covariation. CVPR 2021.
>
> We wish that our response has addressed your concerns, and turns your assessment to the positive side. If you have any questions, please feel free to let us know during the rebuttal window. We appreciate your suggestions and comments! Thank you!

---

> > ### Comment · Reviewer_BwFB · 2021-08-23
> > **Thank you for your reply**
> >
> > Dear authors,
> >
> > thank you for taking the time to read and consider my feedback and thank you for clarifying your results and methods.
> > After reading your rebuttal as well as the responses from other reviewers I am raising my score to 6.
> >
> > If you'll allow me a final suggestion, I think the content of your responses to Q1 and Q1 should be added to the discussion in the paper. It helps put your work in context and clarifies some of the choices you made. I hope you will consider this when preparing your final draft.
> >
> > Thank you again for taking the time to address my suggestions,
> > All the best!

---

> ### Author Response · Authors · 2021-08-22
> **Looking forward to your post-rebuttal feedback**
>
> Dear Reviewer BwFB,
>
> Thanks again for your insightful suggestions and comments. As the deadline for discussion is approaching, we are happy to provide any additional clarifications that you may need.
>
> In our previous response, we have carefully studied your comments and made detailed responses summarized below:
>
> * Provided the definition and detailed analysis of interpretability, and qualitative comparison with counterparts [19].
> * Conducted additional experiments to study the impact of pre-trained object detection modules on data efficiency, and verify the data efficiency of our framework.
> * Discussed the two additional datasets (CATER [23] and ACRE [B]) used in Object-Based Attention [19] and our dataset settings for experiments.
>
> We hope that the provided new experiments and additional explanations about interpretability/datasets have convinced you of the merits of our submission.
>
> Please do not hesitate to contact us if there are other clarifications or experiments we can offer. Thanks!
>
> Thank you for your time!
>
> Best,
> Authors

---

### Official Review · Reviewer_etLB · 2021-07-14

**Rating:** 7
**Confidence:** 4

**Summary:**

- The paper presents a composite model for visual question answering in scenarios which require dynamic visual reasoning.
- The model pipeline consists of a visual perception module which extracts object-centric trajectories from an input video, a concept learner which parses the natural language QA pair into a set of neurosymbolic concept vectors, a differentiable physics model which can simulate the evolution of rigid body trajectories and a neurosymbolic program executor which computes or selects the answer for a given input question using the intermediate concept and trajectory features.
- As a contribution over prior art in neurosymbolic visual reasoning, the authors propose the utilisation of a differentiable physics module for accurate trajectory prediction based on physics parameters which are inferred from the video input sample. Since explicit physical properties like mass, friction and restitution are inferred in the model, the computed output is very amenable to human interpretation.
- The proposed approach is comprehensively evaluated on the synthetic CLEVRER and realistic Real-billiard benchmarks and compared against related neurosymbolic and end-to-end VQA models. It demonstrates remarkable boosts in data efficiency on CLEVRER, sets a new state-of-the-art performance on counterfactual questions and significantly outperforms prior art in a few-shot setup where generalisation to modified physical attributes like 'heavier' or 'lighter' is required.

**Limitations And Societal Impact:**

- The authors have provided a comprehensive list of failure modes of their model in the 'Failure Analysis' paragraph of section 4.2.
- The foreseeable societal impact of the method proposed has been discussed adequately in the Broader Impact section.

**Main Review:**

TL;DR: The paper proposes a clean neurosymbolic pipeline approach for visual question answering involving dynamical reasoning. The introduction of a differential physics module seeded from object collisions massively increases data efficiency and few-shot generalization compared to prior art. The approach has been clearly presented and comprehensively evaluated on two established benchmarks featuring synthetic and real videos depicting rigid body collisions. All claims are corroborated by experiments, the limitations of the approach are outlined. A very good paper which is recommended for acceptance. It would be nice, if the authors provided code and data used in their experiments upon publication.


### Pros and Cons
- Pros
  - The utilisation of a differentiable physics engine affords massive boost in data efficiency during learning and improves performance, especially in counterfactual QAs.
  - The approach is comprehensively evaluated on synthetic and real data; a small-scale dataset extension has been collected to evaluate the model for few-shot generalization to new physical concepts like 'heavier' or 'lighter'.
- Cons
  - The paper only provides sparse details on concept embedding and neurosymbolic program execution -> Maybe some details about those parts of the pipeline should be added to appendix to make the paper more self-contained for the unfamiliar reader?
  - The proposed approach relies heavily on collision events and exact modelling of physics surrounding it.
  - There is no clear evidence presented that the model performance delta over end-to-end models can hold in high-data regimes.
  - The small scale dataset for generalization experiments should have been provided as supplementary material or reported in the appendix to gauge the data distribution of the QAs collected.

### Originality
- Introducing prior knowledge about rigid body physics (especially collision events) affords massive data efficiency boosts on the physical reasoning side since parameters can be fitted well per sample input video and accurate forward prediction can be achieved.
- While this observation about the benefits of prior knowledge injection into a QA model is not novel per se, the comprehensive evaluation of the approach in low-data regimes and counterfactual scenarios is a worthwhile contribution to this field.

### Quality
- typos, formatting errors:
  - l. 61: **a** motion heuristic
  - l. 140: **physics** engine
  - l. 152, 153, 197: coordinate **frame**
  - l. 202: **is** acting on
  - l. 218-219: from ~~less~~ fewer to more frames
  - l. 231: program **outputs** the number
  - l. 263: follow the experimental setting in [75, 16] ~~that~~ using a
  - l. 267, 282: object attribute **supervision**

### Clarity
- The paper is consistently well written and the model pipeline is presented in a clean and understandable way.
- All claims, especially w.r.t. data efficiency and few-shot generalization are sufficiently corroborated in the experiments section.
- There are a few open questions, please clarify during the discussion phase:
  - l. 155, 162: Is `b^n_t` a coordinate since it is used for distance computation?
  - l. 171: How is the concept space being trained?
  - ll. 173-174: What happens to unassigned concepts which aren't visually grounded?
  - How are trajectory features being computed?

### Significance
- Physical reasoning from visual input is highly relevant for visual understanding and applications such as robotic manipulation or navigation, doubly so when paired with natural language descriptions and data-efficient training and inference techniques.
- The neuro-symbolic grounding and forward predictive aspects of the proposed VRDP model make it highly relevant for applications in object-centric scene inference or model-based planning.

### Remarks
- The approach relies heavily on collision events to estimate sample-dependent physics parameters. -> How could this be done in collision-free videos?
- The physics engine only helps to generalize physical reasoning, but embeddings of visual and textual content can mainly be improved by data (as recently shown in multi-modal transformers). -> Are there forseeable cases where more data wouldn't be beneficial to better trajectory prediction, e.g. where [19] would saturate but VRDP keeps improving?
- How does the model fare in very crowded scenes where many enter/exit events occur? Which enhancements to the visual perception module and dynamics heuristic would be needed in such cases?

**Time Spent Reviewing:**

8h

---

> ### Author Response · Authors · 2021-08-10
> **# Response to Reviewer #etLB**
>
> Thank you for the positive comments and insightful suggestions.
>
> **Q1. [Provide code and data upon publication.]** \
> Thanks. We have released our current code and data in https://www.dropbox.com/sh/cetn5jgxd7s22ac/AAAV71alEXaaOLb4gJeE2E50a?dl=0. We will further update the code shortly.
>
>
> **Q2. [How to deal with collision-free videos.]** \
> Our framework is general in rigid body dynamics and works well in collision-free videos. In fact, collision-free video reasoning is less challenging because all objects have only linear motions. In this simplified case, our approach can easily and accurately estimate the velocity and friction of objects and perform reasoning based only on these non-collision physical parameters.
>
> We also agree that we cannot estimate the collision-based physical parameters like mass and restitution in collision-free videos. However, we are not expected to estimate that because even humans can not answer collision-based questions from collision-free videos.
>
>
> **Q3. [Performance compared to end-to-end models [19] under high-data regimes.]** \
> Good question. Data efficiency is one of the main purposes of this work as it is important to reasoning researches, where the cases cannot be fully traversed. We believe that reasoning from few samples is the key to build human intelligence.
>
> In high-data regimes, no clear evidence can indeed show our explicit physics model consistently outperforms implicit end-to-end models. But intuitively, long-term predictions and counterfactual reasoning are challenging for end-to-end models because these two tasks require the model to have the ability to imagine and simulate. End-to-end models may learn some "shortcuts" or "bias" rather than precise imagination as our physics simulator.
>
> End-to-end models [19] typically fail in some generalized settings, while our VRDP model works well. An examplar case is that the testing videos contain novel compositionality of visual attributes and dramatically changed physical parameters. For example, the testing video in Section 4.2 has a physical parameter (larger mass value) that is much different from those of the training videos. Our model can identify and learn this difference from the given test video, while the end-to-end model [19] does not.
>
>
> **Q4. [Report the small scale dataset in the appendix.]** \
> Good suggestion. We will add more details about neuro-symbolic programs and our collected dataset in the revision.
>
>
> **Q5. [Typos and formatting errors.]** \
> Thanks for the careful reading. We will fix all the typos in the revision.
>
>
> **Q6.1. [Is $b^n_t$ a coordinate since it is used for distance computation?]** \
> Yes. It is the pixel coordinate of the detected central location of objects (4-dimensional). Specifically, the object location of the $n_{th}$ object $b^n_t=[x^{n^\text{2D}}_t, y^{n^\text{2D}}_t, x^{n^\text{BEV}}_t, y^{n^\text{BEV}}_t] \in \mathbb{R}^{4}$ at frame $t$, $(x^{n^\text{2D}}_t, y^{n^\text{2D}}_t)$ denotes the normalized object bounding box center in the image coordinate, and $(x^{n^\text{BEV}}_t, y^{n^\text{BEV}}_t)$ denotes the projected bird's-eye view (BEV) location in the BEV coordinate using the calibrated camera matrix.
>
>
> **Q6.2. [Training of the concept space.]** \
> The concept space is learned along the neuro-symbolic reasoning process and supervised by question-answer pairs. We measure the cosine distances between the visual feature and concept vectors to determine the confidence score of an object attribute. In this way, the program executor has a fully differentiable design w.r.t. the visual representations and the concept representations, which supports gradient-based optimization.
>
> The training process is shown in lines 135-144. During training, we first learn the object and event concept spaces with descriptive and explanatory questions that do not require dynamic predictions (without the physics model). After the object properties are grounded, our physics model can help the concept learner finetune event concepts from the counterfactual and predictive programs.
>
>
> **Q6.3. [What happens to unassigned concepts which aren't visually grounded?]** \
> Currently, our method is based on a closed set of language concepts. All concepts used in the dataset are pre-defined and seamlessly integrated into neuro-symbolic programs. There are no unassigned concepts, i.e., concepts like "color and shape", "enter and moving", and "collision" are grounded with object features, trajectory features, and interactive features, respectively.
>
> Specifically, all concepts in the neuro-symbolic program are represented by embeddings which are then used for distance measuring with the corresponding visual features. For example, the confidence score of whether the $n^{th}$ object is a cube can be represented by $[\mathrm{cos}(\mathcal{P}(F_a^{n}), e_\text{cube}) -\mu]/\sigma$ (lines 176-179), where $e_\text{cube}$ is the embedding of the concept cube. Exploring unknown concepts is a good topic in our future work.
>
>
> **Q6.4. [Computation of the trajectory feature.]** \
> As shown in lines 155-158, the trajectory feature $F_l=\\{ b_t \\}^T_{t=1}$ is represented by object locations on the image and BEV planes in each video frame (4NT-dimensional where N is the number of objects and T is the number of frames).
>
>
> **Q7. [Enhancements to handle very crowded scenes with many enter/exit events.]** \
> As shown in our failure analysis (lines 326-334), the optimization of physical parameters becomes difficult if there are dense collisions or insufficient trajectories. In the case that many enter/exit events occur, the biggest problem is insufficient frames to learn the initial velocity $v_0$. In our current implementation, we count an object in the observed trajectory for physical optimization only when most of the area of the object is in the image plane. This is because if an object is on the boundary of the image plane (truncated), its central location might be perceived inaccurately.
>
> We can handle this corner case with a more powerful perception module. We may collect enough data that objects are located on the boundary of the image plane and train a better perception model to learn accurate object locations even if truncated. In this way, all truncated frames can be used as trajectories for physical optimization, which will increase the robustness of our model significantly. We can also divide the video in spatial and temporal, and optimize the short-term motion of a few objects each time, to reduce the difficulty of optimization.

---

> > ### Comment · Reviewer_etLB · 2021-08-23
> > **Thank you for the clarifications!**
> >
> > Dear authors,
> >
> > Thank you very much for responding in such a detailed fashion! The additional experiments (requested by other reviewers) and preview of the source code have also just added to my overall impression that this work is of very high quality and should definitely be recommended for acceptance. If space permits, I would very much appreciate if you added your responses to `Q6.1` and `Q6.4` to the final manuscript. Also, the clarification provided in `Q6.3` would be worth mentioning in the discussion section to encourage future work in this direction.
> >
> > Thanks again for your time, effort and diligence put into this rebuttal!

---

### Official Review · Reviewer_Dj3E · 2021-07-15

**Rating:** 6
**Confidence:** 4

**Summary:**

The paper tackles the problem of dynamic visual reasoning by using a modular unit that is composed of three components: visual perception, concept learner, and physics simulator. This system is capable of performing system identification from visual and language modalities with assistance from a physics model. Having access to the physics model, it can produce imagined future trajectories for reasoning about possible future outcomes. This gave them an advantage of outperforming competitive methods, especially in counterfactual tasks.


**Limitations And Societal Impact:**

It has been addressed in the paper.

**Main Review:**

Overall:

It’s a well-written paper, that formulates a problem, proposes a solution, and does a reasonable analysis of the results. The results are strong and competitive especially in the scenarios such as counterfactual where they do argue why their method should perform better.

While the current system could have a great many applications in a controlled robotics environments, the system looks a bit brittle and specialized and it’s not clear how this system could be generalized to other datasets and applications.

Details:

Is understanding physical laws and having a physics engine a necessary condition for physics reasoning? The intuitive physics in humans suggests otherwise. While it’s very reasonable to have a physical simulator for robotics applications, it might not necessarily be a requirement for AGI.

Would this model be capable to expand more complicated physical models, such as complicated contact dynamics that happens in manipulation?

By just reading the main text alone the mechanism of “Symbolic programs” and “Symbolic excitation” are not clear. While the authors cite relevant work and have a section in the appendix, the text needs to be self-sufficient and it would be great to have a section to explain those in more detail.

How does the computation efficiency of VRDP compare to the other methods?

**Time Spent Reviewing:**

4

---

> ### Author Response · Authors · 2021-08-10
> **# Response to Reviewer #Dj3E**
>
> Thank you for the insightful comments and suggestions.
>
> **Q1. [The necessity of physical laws and engines for AGI.]** \
> Interesting question. We agree that the intuitive physics in humans can reason about the physical world without explicit physical laws. However, it is still very challenging for a machine to understand the physical world. Considering the full promise of machine intelligence has yet to be realized, we believe it will be promising to build physics knowledge into the learning framework and promote understanding of the physical world. In our work, we showcase that integrating a differentiable physics engine into a neuro-symbolic framework can achieve more generalized and data-efficient physical reasoning ability.
>
>
> **Q2. [Expand more complicated physical models, e.g., contact dynamics in manipulation.]** \
> Good question. This work interprets dynamic visual reasoning into a general framework containing visual perception, concept learning, and differentiable physics simulation. The framework is general and can be extended to more complicated scenes by updating the submodules. For example, as shown in section 4.2 (evaluation of generalizability), our model performs counterfactual reasoning with a new word "heavier" easily by updating the concept in the concept learner. We have also shown that our differentiable physics simulator can be easily extended to objects with complex shapes. Please see our response to Q1 of reviewer #Qwxn and the experimental results in https://www.dropbox.com/s/2hu8m0084i1qres/VRDP_on_Complicated_Shapes.mp4.
>
> Our current rigid-body physics engines do not support soft-body dynamics and robotic manipulation, but we can flexibly integrate new components or engines  (e.g., particle-based physics engines [A] and end-effector control [B]) into this neuro-symbolic framework. We leave them as our future work.
>
> [A] Li, Yunzhu, et al. "Learning particle dynamics for manipulating rigid bodies, deformable objects, and fluids." ICLR 2019.
> [B] Wilson, William J., et al. "Relative end-effector control using cartesian position based visual servoing." IEEE Transactions on Robotics and Automation, 1996.
>
>
> **Q3. [Explain symbolic programs and excitation in more detail.]** \
> Thanks for your suggestion. We will reorganize and explain the symbolic mechanism in detail in the revision.
>
>
> **Q4. [The computation efficiency of VRDP.]** \
> VRDP contains three modules: a visual perception module, a concept learner, and a physics model. Following common practice in [75, 16], the visual module is a Faster-RCNN model and the concept learner contains a seq2seq network. Considering that all counterparts, including the end-to-end method [19], need to extract visual representations from a pretrained perception model (Faster-RCNN or MONet), the computational costs of our visual module and concept learner/program executor are similar to existing works [75, 16, 19].
>
> In the dynamic prediction part, our differentiable physics engine estimates the physical parameters from observed trajectories then perform physical simulation. This process consumes dozens of seconds on CPUs in our current implementation, which costs more time than the counterparts that use a deep model for dynamic prediction on GPUs. However, our optimization and simulation processes can be accelerated in real-world applications by parallel computing on GPUs. For example, the benchmark [C] shows that rigid-body physics simulation in state-of-the-art works can be accelerated to two to three orders of magnitude faster than in real-time. We leave it as our future work.
>
> [C] Erez, Tom, Yuval Tassa, and Emanuel Todorov. "Simulation tools for model-based robotics: Comparison of bullet, havok, mujoco, ode and physx." IEEE international conference on robotics and automation (ICRA), 2015.

---

### Official Review · Reviewer_Qwxn · 2021-07-17

**Rating:** 7
**Confidence:** 5

**Summary:**

This paper proposes to do visual question answering by learning a differentiable physics model from video. The pipeline is divided to several modular and interpretable parts. Firstly, a visual perception module parse the input video to several objects’ features. Secondly, a concept learner parse the input questions to a set of programs and grounded concepts. The output of the above two modules are used to estimate the objects’ physical parameters and fed to a differentiable physical engine, which is used to do future trajectory predictions. Finally, a symbolic execution part is used to output answer given the future predictions.

**Limitations And Societal Impact:**

I didn't find any potential negative societal impact of this work.

**Main Review:**

This paper presents a new way to do visual reasoning by incorporating a differentiable physics engine to make future prediction. I think this is a novel introduction of an advanced method to the VQA field. And as the authors claim, the advantage of modularized design is that the intermediate outputs are interpretable so it’s easier to identify the reason of failure. More importantly, thanks to the differentiable simulator, the method does not lose the ability to end-to-end optimization.

In addition, incorporating the differentiable physics engine can greatly improve the sample efficiency due to the introduction of human prior. The author verifies this idea by showing the proposed method is several times better than previous state-of-the-art under the 20% training data regime and the few-shot learning setting.

The paper is also well-written and very easy to follow.

Questions:
1. Have the authors considered how to apply this to more complex environments? For example, most of the objects in CLEVERER are blocks and spheres and the current physics simulation model assume to be a mass-point. How would the proposed method be extended to more complex objects’ shapes?
2. Have the authors considered another ablation experiments: using learned dynamics prediction model to replace the differentiable physics simulator? For example, what if we use the models from [51, 59] to do simulation?

**Time Spent Reviewing:**

4.0 hours

---

> ### Author Response · Authors · 2021-08-10
> **Response to Reviewer #Qwxn**
>
> Thank you for the positive comments and insightful suggestions.
>
> **Q1. [The current physics simulation assumes the object to be a mass-point. How would it be extended to more complex objects’ shapes?]** \
> Sorry for the confusion. The objects considered in this paper are not mass points because they have their own radius, shape, and coefficient of friction. Colliding with different regions of these objects results in different dynamic changes, e.g., an object will begin rotating if it is subjected to a collision at the point not directed to the center of mass (see the rotation of cubes in the video of supplementary materials). We also show the circle-rectangle collision detection algorithm in Figure 1 of supplementary materials.
>
> Since our concept learner and symbolic representation can be easily extended to support new concepts or shapes, it is no need to worry that our framework will fail when more complex shapes are added. The challenge for supporting more complex objects is mainly in collision detection, which is purely a geometry problem (giving two shapes and guessing when they collide) not a physics simulation problem, and mature solutions [A] exist. Therefore, with the rigid-body dynamics model unified for different shapes, our framework is general and can be easily extended to objects with complex shapes.
>
> To demonstrate that, we add new experiments with an irregularly shaped object (i.e., a bottle). The qualitative results in https://www.dropbox.com/s/2hu8m0084i1qres/VRDP_on_Complicated_Shapes.mp4 show that our differentiable physics model can reveal the physics of objects with complex shapes and align the simulation with ground-truth observations as closely as possible. The precise physical simulation then helps solve downstream question-answering problems. In this experiment, we add a new shape concept "bottle" and update our collision detection algorithm between bottles and other objects, while our rigid-body dynamics model remains unchanged. Specifically, we use the fourth-order Bezier curve to describe the shape of the bottle and calculate the distance from a point to the curve for collision detection.
>
> [A] Erwin Coumans and Yunfei Bai. "PyBullet, a Python module for physics simulation for games, robotics and machine learning", 2016.
>
>
> **Q2. [Quantitative comparison between differentiable physics and dynamics prediction models [51, 59].]** \
> Thanks for the suggestion. We conducted comparative experiments of the framework with our differentiable physics simulator against that with the dynamic prediction model PropNet [51] on the CLEVRER dataset. With the same visual representation and neuro-symbolic programs, our physics model outperforms PropNet [51] on all four types of QAs. We observed that:
> * With the PropNet [51] replacing our differentiable physics model, the per-question QA accuracies on predictive, counterfactual, descriptive, and explanatory questions are decreased by 10.1%, 34.7%, 2.2%, and 8.8%, respectively.
> * PropNet achieves comparable performance to ours on descriptive questions.
> * Our physics simulator can better model the causal events such as collision, thus perform better on explanatory questions.
> * Our physics simulator outperforms PropNet by large margins on long-term predictions and counterfactual events.
>
> We can conclude that dynamics prediction models [51] have basic reasoning and short-term prediction abilities. Still, it is difficult for them to deal with complex reasoning problems, such as accurate collision detection, long-term and counterfactual predictions.

---

### Author Response · Authors · 2021-08-10
**General Response: Contributions and New Experiments**

We sincerely appreciate all reviewers’ time and efforts in reviewing our paper. We are glad to find that reviewers generally recognized our contributions:
* **Model.** Introducing a new way and cool idea of combining visual perception, concept learning, and physics simulation into a reasoning system [Qwxn, BwFB]; Achieving better interpretability, data efficiency, and few-shot generalization as explicit physical properties are inferred in the model [Qwxn, etLB, BwFB].
* **Experiments.** Corroborating all claims with reasonable analysis and comprehensive evaluation [Dj3E, etLB]; Showing remarkable and promising results, especially in counterfactual scenarios [Dj3E, etLB, BwFB].
* **Writing.** Having the paper clearly presented and easy to follow [Qwxn, Dj3E, etLB, BwFB].

And we also thank all reviewers for their insightful and constructive suggestions, which help a lot in further improving our paper. In addition to the pointwise responses below, we summarize supporting experiments added in the rebuttal according to reviewers’ suggestions.

**New Experiments**
* Extending our model to more complex objects’ shapes [Qwxn, Dj3E] (https://www.dropbox.com/s/2hu8m0084i1qres/VRDP_on_Complicated_Shapes.mp4);
* Comparative experiments using the dynamic prediction model PropNet to replace the differentiable physics simulator in our framework [Qwxn];
* Ablative experiments on the impact of pre-trained visual detectors on data efficiency [BwFB];
* Release the code and data [etLB] (https://www.dropbox.com/sh/cetn5jgxd7s22ac/AAAV71alEXaaOLb4gJeE2E50a?dl=0).

We hope our pointwise responses below could clarify all reviewers’ confusion and alleviate all concerns. We thank all reviewers’ time again.

---

### Author Response · Authors · 2021-09-01
**Summary of our rebuttal and discussion**

We sincerely appreciate all reviewers’ and ACs’ time and efforts in reviewing our paper. We truly thank you all for the insightful and constructive suggestions, which helped further improve our paper. We genuinely appreciate the positive 6-7-6-7 evaluation from reviewers BwFB, etLB, Dj3E, and Qwxn.

Here is a summary of our updates:

* [Additional Experiments] As suggested by reviewers BwFB, Dj3E, and Qwxn, we conduct extra experiments on pre-trained visual detectors, other dynamic prediction models, and more complex objects’ shapes. All additional results consistently validate the effectiveness of our proposal.
* [Code] We released our code and data as suggested by reviewer etLB.
* [Writing] We owe many thanks to reviewer etLB’s extremely helpful writing suggestions. All improved manuscript parts, together with other constructive discussions with reviewers BwFB, Dj3E, and Qwxn, will be delivered in our final version.

We really thank all reviewers’ and ACs’ time and efforts again.

Best wishes, \
Authors

---

### Decision · Program_Chairs · 2021-09-27

**Decision:**

Accept (Poster)

**Comment:**

The paper proposes a new task and method for dynamic visual question answering based on neuro-symbolic reasoning and the usage of a differentiable physics engine. It has received reviews from four experts, who appreciated the new formulation with the differentiable engine (and its effect on counterfactual tasks), the modular and interpretable design, and a well written and presented paper.

Minor issues were raised on the model-based nature of the method --- the advantage of having a physics engine is also a shortcoming, since the method requires a model of the physical processes, and on evaluation, comparisons.

The authors' responses were convincing, in particular some requested experiments (eg re: pre-training on objects) where appreciated by the reviewers, and a consensus for acceptance emerged.

The AC recommends acceptance.